# TEST-TIME SCALING OF DIFFUSIONS WITH FLOW MAPS

## ABSTRACT

A common recipe to improve diffusion models at test-time so that samples score highly against a user-specified reward is to introduce the gradient of the reward into the dynamics of the diffusion itself. This procedure is often ill posed, as user-specified rewards are usually only well defined on the data distribution at the end of generation. While common workarounds to this problem are to use a denoiser to estimate what a sample would have been at the end of generation, we propose a simple solution to this problem by working directly with a flow map. By exploiting a relationship between the flow map and velocity field governing the instantaneous transport, we construct an algorithm, Flow Map Trajectory Tilting (FMTT), which provably performs better ascent on the reward than standard test-time methods involving the gradient of the reward. The approach can be used to either perform exact sampling via importance weighting or principled search that identifies local maximizers of the reward-tilted distribution. We demonstrate the efficacy of our approach against other lookahead techniques, and show how the flow map enables engagement with complicated reward functions that make possible new forms of image editing, e.g. by interfacing with vision language models.

| ChatGPT | Gemini 🍌 | FLUX.1 [dev] | FLUX + FMTT (Ours) |
|---|---|---|---|

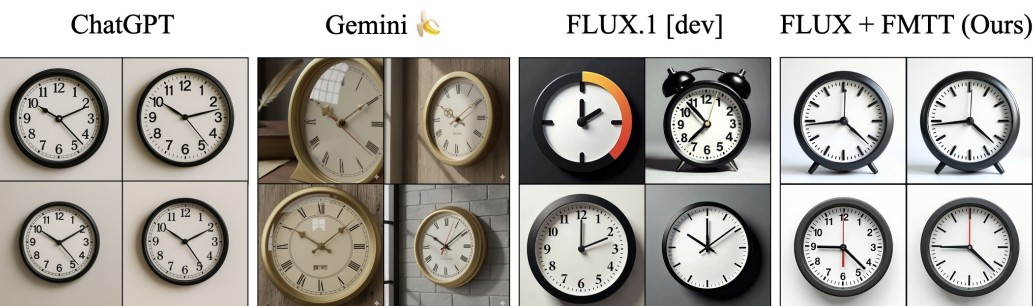

*Prompt: "An analog clock showing **exactly 4:45**"*

**Figure 1:** Test-time search can overcome model biases and reliably sample from regions of the distribution (e.g., precise clock times) that baselines fail to capture.

## 1 INTRODUCTION

Large scale foundation models built out of diffusions (Ho et al., 2020; Song et al., 2020) or flow-based transport (Lipman et al., 2022; Albergo & Vanden-Eijnden, 2022; Albergo et al., 2023; Liu et al., 2022) have become highly successful tools across computer vision and scientific domains. In this paradigm, performing generation amounts to numerically solving an ordinary or stochastic differential equation (ODE/SDE), the coefficients of which are learned neural networks. An active area of current research is how to best *adapt* these dynamical equations at inference time to extract samples from the model that align well with a user-specified reward. For example, as shown in Figure 1, a user may want to generate an image of a clock with a precise time displayed on it, which is often generated inaccurately without suitable adaptation of the generative process. These approaches, often collectively referred to as *guidance*, do not require additional re-training and as a result are orthogonal to the class of fine-tuning methods, which instead attempt to adjust the model itself via an additional learning procedure to modify the quality of generated samples.

While guidance-based approaches can often be made to work well in practice, most methods are somewhat *ad-hoc*, and proceed by postulating a term that may drive the generative equations towards the desired goal. To this end, a common approach is to incorporate the gradient of the reward model, which imposes a gradient ascent-like structure on the reward throughout generation. Despite the intuitive appeal of this approach, typical rewards are defined only at the terminal point of generation – i.e., over a clean image – rather than over the entire generative process. This creates a need to "predict" where the current trajectory will land, in principle necessitating an expensive additional differential equation solve per step of generation. To avoid the associated computational expense of this nested solve, common practice is to employ a heuristic approximation of the terminal point, such as leveraging a one-step denoiser that can be derived from a learned score or flow-based model.

In this paper, we revisit the reward guidance problem from the perspective of flow maps, a recently-introduced methodology for flow-based generative modeling that learns the solution operator of a probability flow ODE directly rather than the associated drift (Boffi et al., 2024; 2025; Sabour et al., 2025; Geng et al., 2025). By leveraging a simple identity of the flow map, we show that an implicit flow can be used to define a reward-guided generative process as in the case of standard flow-based models. With access to the flow map in addition to the implicit flow, we can predict the terminal point of a trajectory in a single or a few function evaluations, vastly improving the prediction relative to denoiser-based techniques and leading to significantly improved optimization of the reward. In addition, we highlight how to incorporate time-dependent weights throughout the generative process to account for the gradient ascent's failure to equilibrate on the timescale of generation, leading to several new and effective ways to sample high-reward outputs.

**Contributions.** (*i*) We introduce Flow Map Trajectory Tilting (FMTT), a principled inference time adaptation procedure for flow maps that effectively uses their look-ahead capabilities to accurately incorporate learned and complex reward functions in Monte Carlo and search algorithms. (*ii*) Using conditions that characterize the flow map, we show that the importance weights for this Jarzynski/SMC scheme reduce to a remarkably simple formula. Our approach is theoretically grounded in controlling the *thermodynamic length* of the process over baselines, a measure of the efficiency of the guidance in sampling the tilted distribution. (*iii*) We empirically show that FMTT has favorable test-time scaling characteristics that outperform standard ways of embedding rewards into diffusion sampling setups. (*iv*) To our knowledge, we demonstrate the first successful use of pretrained vision-language models (VLMs) as reward functions for test-time scaling, allowing rewards to be specified entirely in natural language. We further show that the flow map is crucial for their success, substantially boosting the effectiveness of the search process when using these rewards.

## 1.1 RELATED WORK

**Flows and diffusions.** Diffusion models (Song et al., 2020; Ho et al., 2020) and flow models (Lipman et al., 2022; Albergo & Vanden-Eijnden, 2022; Liu et al., 2022) are the backbone of efficient, state-of-the-art generative model for continuous data. They are learned by regressing the coefficients that appear in ordinary or stochastic differential equations that fulfill the transport of samples from one distribution to samples from another. Dual to the instantaneous picture of transport is the flow map (Song et al., 2023; Kim et al., 2024; Boffi et al., 2024; Geng et al., 2025; Sabour et al., 2025; Boffi et al., 2025), in which we learn not the coefficients in a differential equation that needs to be integrated, but the arbitrary integrator itself. This enables few-step sampling. Our approach in this paper is to combine these perspectives to modify diffusions using the flow map.

**Test-time scaling for diffusions.** Test-time scaling in diffusions refers to the line of work that tradeoff compute at inference time to improve the performance of a model or align the generation with a user specified reward (Ma et al., 2025). Certain works use the denoiser associated with the score model to perform this look-ahead on the dynamics (Wu et al., 2023a; Singhal et al., 2025; Zhang et al., 2025) or do not use any look-ahead at all (Mousavi-Hosseini et al., 2025; Skreta et al., 2025). However, as discussed later, there is little signal from the denoiser at early times in the generative trajectory. Other works rely on Monte Carlo search algorithms (Lee et al., 2025; Ramesh & Mardani, 2025), which monotonically increase the reward but reduce sample diversity. As we will see, many of these approaches are compatible with the flow map approach presented here.

## 2 METHODOLOGY

We consider the task of generative modeling via continuous-time flow maps, wherein samples $x_0 \in \mathbb{R}^d$ from a base distribution with probability density function (PDF) $\rho_0$ are mapped via a diffeomorphism to samples $x_1$ from the target PDF $\rho_1$ known through empirical data. From there, we will detail how the instantaneous dynamics of this map can be directly adapted (without retraining) to sample a **tilted distribution** favoring a reward, i.e. to sample $\hat{\rho}_1(x) = \rho_1 e^{r(x)+\hat{F}}$ where $r(x)$ is a user specified reward function and $\hat{F} = -\ln \int_{\mathbb{R}^d} \rho_1(x) e^{r(x)} dx$ is a normalization factor.

### 2.1 BACKGROUND ON DYNAMICAL GENERATIVE MODELING

An effective means of instantiating the transport from the base PDF $\rho_0$ to the target PDF $\rho_1$ relies on formulating it as the solution to an ordinary differential equation (ODE) of the form

$$\dot{x}_t = b_t(x_t) \qquad x_{t=0} \sim \rho_{t=0}, \tag{1}$$

where $b_t : [0,1] \times \mathbb{R}^d \to \mathbb{R}^d$ is a velocity field that governs the transport and is adjusted so that the solutions to the ODE (1) satisfy $x_{t=1} \sim \rho_1$. Since the time dependent PDF $\rho_t(x)$ of the solutions to (1) at time $t$ satisfies the continuity equation

$$\partial_t \rho_t = -\nabla \cdot (b_t \rho_t) \qquad \rho_{t=0} = \rho_0 \tag{2}$$

this requirement on $b_t$ implies that the solution to (2) is such that $\rho_{t=1} = \rho_1$.

Associated with these dynamics is the two-time **flow map** $X_{s,t} : [0,1]^2 \times \mathbb{R}^d \to \mathbb{R}^d$, which satisfies

$$X_{s,t}(x_s) = x_t \qquad \forall s, t \in [0,1]. \tag{3}$$

That is, the map jumps along solutions of (1) from time $s$ to time $t$. Notably, if $s = 0$ and $t = 1$, we could produce a sample under $\rho_1$ in a single step, though we have the freedom to use more if we so choose. This property, and the relation between the flow map and $b_t$ will be exploited below to devise a principled adaptation procedure for $X_{s,t}$. Importantly, the flow map satisfies the Eulerian equation

$$\partial_s X_{s,t}(x) + b_s(x) \cdot \nabla X_{s,t}(x) = 0, \tag{4}$$

which will play a role in simplifying our analysis later. Equation (4) can be obtained by taking the total derivative of (3) with respect to $s$, using the ODE (1), and evaluating the result at $x_s = x$.

**Stochastic Interpolants.** One way to instantiate the generative models above is to construct a PDF $\rho_t$ that connects $\rho_0$ to $\rho_1$ and then learn the associated the velocity field $b_t$ that gives rise to this evolution. A common strategy to construct such a path and regress $b_t$ is that of stochastic interpolants (Albergo & Vanden-Eijnden, 2022; Albergo et al., 2023; Lipman et al., 2022; Liu et al., 2022), in which $\rho_t$ is defined as the law of the stochastic process $I_t(x_0, x_1) = \alpha_t x_0 + \beta_t x_1$ with $(x_0, x_1) \sim \rho(x_0, x_1)$, where $\rho(x_0, x_1)$ is some coupling from which $x_0, x_1$ are drawn that marginalizes onto $\rho_0, \rho_1$ and $\alpha_t, \beta_t$ are scalar coefficients that satisfy $\alpha_0 = \beta_1 = 1$ and $\alpha_1 = \beta_0 = 0$. A common choice is to use $\alpha_t = 1 - t$ and $\beta_t = t$, which we will use throughout for simplicity. Importantly, using these coefficients, the law of this process satisfies (2) with the velocity field

$$b_t(x) = \mathbb{E}[\dot{I}_t | I_t = x] = \mathbb{E}[x_1 | I_t] - \mathbb{E}[x_0 | I_t], \tag{5}$$

where we used $\dot{I}_t = x_1 - x_0$ and $\mathbb{E}[\cdot | I_t = x]$ denotes expectation over $\rho(x_0, x_1)$ conditional on $I_t = x$. By Stein's identity, the score is given by $s_t(x) = \nabla \log \rho_t(x) = -\frac{1}{1-t} \mathbb{E}[x_0 | I_t = x]$, and using $x = \mathbb{E}[I_t | I_t = x] = (1 - t)\mathbb{E}[x_0 | I_t] + t\mathbb{E}[x_1 | I_t]$, it can be expressed in terms of $b_t$ as

$$s_t(x) = (tb_t(x) - x)(1 - t)^{-1}. \tag{6}$$

The velocity field $b_t(x)$ is also the minimizer of a simple quadratic objective (Lipman et al., 2022; Albergo & Vanden-Eijnden, 2022) which, once learned, can be translated into a function for the score via (6). Using the score, the deterministic ODE can be converted to a stochastic dynamics

$$dx_t = [b_t(x_t) + \epsilon_t s_t(x_t)] \, dt + \sqrt{2\epsilon_t} dW_t \tag{7}$$

where $\epsilon_t \geq 0$ is an arbitrarily tunable diffusion coefficient and $dW_t$ is an incremental Brownian motion (Albergo et al., 2023). The solutions to (7) sample the same PDF $\rho_t$ as (1), as can be seen from the fact that the PDF of (7) satisfies the Fokker-Planck equation

$$\partial_t \rho_t = -\nabla \cdot (b_t \rho_t) + \epsilon_t \nabla \cdot [-s_t \rho_t + \nabla \rho_t] \tag{8}$$

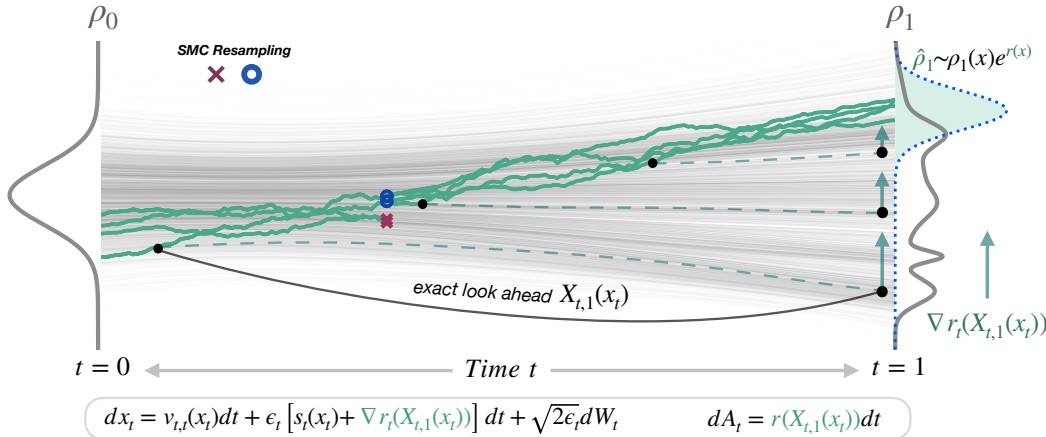

**Figure 2:** Schematic overview of test-time adaptation of diffusions with flow map tilting. Using the look-ahead map $X_{t,1}(x_t)$ in the diffusion inside the reward, reward information can be principly used through the tilted trajectories (green lines). This allows us to perform better ascent on the reward, and the importance weights $A_t$ take on a remarkably simple form that can be used for both exactly sampling $\hat{\rho}_t$ and search for maximizers of $\hat{\rho}_t$.

which reduces to (2) since $s_t \rho_t = \nabla \rho_t$. The velocity field given in (5) is related to the two-time flow map $X_{s,t}$ via the tangent identity (Kim et al., 2024; Boffi et al., 2025)

$$\lim_{s \to t} \partial_t X_{s,t}(x) = b_t(x), \tag{9}$$

which says that for infinitesimally small steps, the variation of the flow map output in time is characterized by the velocity. To automatically enforce the boundary condition $X_{s,s}(x) = x$, we can express the flow map as

$$X_{s,t}(x) = x + (t - s)v_{s,t}(x), \tag{10}$$

where $v_{s,t}(x) : [0,1]^2 \times \mathbb{R}^d \to \mathbb{R}^d$ is a function of $s$, $t$, and $x$ defined through this relation. Importantly, *the velocity field is accessible directly from the flow map by using* (9) *on* (10):

$$v_{t,t}(x) = b_t(x). \tag{11}$$

As such, training a flow map model instead of just a velocity model gives access to both the drifts in (1) and (7) as well as the any step model, i.e. the ability to *look ahead* our trajectory.

## 2.2 Fixing Inference-time adaptation of diffusions with flow maps

A contemporary question is how best to adapt the SDE (7) to tilt toward samples that score highly against a time-dependent reward $r_t(x)$ satisfying $r_0 = 0$ and $r_{t=1} = r$ so that the time-dependent tilted PDF $\hat{\rho}_t = \rho_t e^{r_t(x) + \hat{F}_t}$ satisfies $\hat{\rho}_{t=0} = \rho_0$ and $\hat{\rho}_{t=1} = \hat{\rho}_1 = \rho_1 e^{r + \hat{F}}$. One may want to *sample exactly* under the tilted PDF $\hat{\rho}_t = \rho_t e^{r_t(x) + \hat{F}_t}$ for scientific applications, or one may want to *track local maximizers* of $\hat{\rho}_t$ for example in image generation procedures. This is useful for ensuring user prompts align with image content.

**Tilting the diffusion.** A natural way to modify the SDE (7) is to add the gradient of the time-dependent reward $r_t$ to the score, i.e. use

$$d\tilde{x}_t = [b_t(\tilde{x}_t) + \epsilon_t s_t(\tilde{x}_t) + \epsilon_t \nabla r_t(\tilde{x}_t)]dt + \sqrt{2\epsilon_t}dW_t, \qquad \tilde{x}_0 \sim \rho_0 \tag{12}$$

To implement this change in practice, however, we face an issue:

*A meaningful $r_t(x)$ is not readily available, as user-specified rewards are usually learned only on the data-distribution, i.e. at time $t = 1$.*

One could think of several solutions to this problem: **Naïve look-ahead:** This amounts to using e.g. $r_t(x) = tr(x)$. Unfortunately, the gradient dynamics from $t\nabla r(x)$ provides no clear signal

at small times when $\tilde{x}_t$ is still far from the region where the reward $r(x)$ is meaningful. **Denoiser look-ahead:** A common workaround for the fact that the reward has no signal for most of the trajectory is to use the *denoiser* $\mathsf{D}_t(x) = \mathbb{E}[x_1 | I_t = x]$ to estimate where the sample would have gone. That is, instead of $r_t(x) = tr(x)$, we could instead use $r_t(x) = tr(\mathsf{D}_t(x))$. This strategy is tractable because the denoiser is readily available from the score. However, this still does not provide useful information early on in the dynamics, as the denoiser is only effective at producing samples close to the data distribution later in the evolution. **Flow map look-ahead:** Intuitively, the above dynamics are better if one works instead with the flow map defined in the previous section. Because the flow allows us to look ahead at any point on the trajectory, e.g. by taking $\tilde{x}_t$ at time $t$ and computing $X_{t,1}(\tilde{x}_t)$, and because the velocity field associated to the flow map is accessible via $\lim_{s \to t} \partial_t X_{s,t}(x) = v_{t,t}(x)$, we can instead sample with the following SDE:

$$d\tilde{x}_t = v_{t,t}(\tilde{x}_t)dt + \epsilon_t \left[ s_t(\tilde{x}_t) + t\nabla_{\tilde{x}_t} r(X_{t,1}(\tilde{x}_t)) \right] dt + \sqrt{2\epsilon_t}dW_t, \qquad \tilde{x}_0 \sim \rho_0 \tag{13}$$

where $r_t(x) = tr(X_{t,1})(x)$ makes use of the *exact look-ahead* to properly evaluate the reward for any $t$, even $t = 0$. The above could be interpreted as a continuous deformation of how the 1-step flow map would evolve under ascent on the reward.

## 2.3 Correcting the dynamics for unbiased sampling

While using the flow map composed with the reward makes possible the precise use of the reward for all times in the diffusion trajectory, in applications where it is important to *exactly* sample the tilted distribution $\hat{\rho}_t$, the dynamics in (13) are not sufficient to fulfill this. This gives rise to the second issue, for any version of $r_t(x)$, with or without the flow map:

*The PDF associated with (12) is not the tilted density $\hat{\rho}_t$.*

To see why this is true, note that we can explicitly compare the PDF $\tilde{\rho}_t$ of $\tilde{x}_t$ to that of $\hat{\rho}_t$. Indeed, the PDF $\tilde{\rho}_t(x)$ of $\tilde{x}_t$ satisfies:

$$\partial_t \tilde{\rho}_t + \nabla \cdot (b_t \tilde{\rho}_t) = \epsilon_t \nabla \cdot ([-s_t - \nabla r_t]\tilde{\rho}_t + \nabla \tilde{\rho}_t), \tag{14}$$

and we can show explicitly that $\hat{\rho}_t$ satisfies a different, imbalanced equation which can be obtained by expanding $\partial_t \hat{\rho}_t + \nabla \cdot (b_t \hat{\rho}_t)$:

$$\partial_t \hat{\rho}_t + \nabla \cdot (b_t \hat{\rho}_t) = \partial_t(e^{r_t + \hat{F}_t}\rho_t) + \nabla \cdot (b_t e^{r_t + \hat{F}_t}\rho_t) = (\partial_t r_t + \partial_t \hat{F}_t)\hat{\rho}_t + b_t \cdot \nabla r_t \hat{\rho}_t \tag{15}$$

where we used the FPE (8) with $\epsilon_t = 0$ to get the last equality. Since $\nabla \hat{\rho}_t = (s_t + \nabla r_t)\hat{\rho}_t$ we can add the diffusion term $\epsilon_t \nabla \cdot (-(s_t + \nabla r_t)\hat{\rho}_t + \nabla \hat{\rho}_t) = 0$ to (15) to arrive at

$$\partial_t \hat{\rho}_t + \nabla \cdot (b_t \hat{\rho}_t) = \epsilon_t \nabla \cdot ((-s_t - \nabla r_t)\hat{\rho}_t + \nabla \hat{\rho}_t) + (b_t \cdot \nabla r_t + \partial_t r_t + \partial_t \hat{F}_t)\hat{\rho}_t. \tag{16}$$

As we can see, the extra term $(b_t \cdot \nabla r_t + \partial_t r_t + \partial_t \hat{F}_t)\hat{\rho}_t$ on the RHS of this equation differentiates it from being the law of (12). Nonetheless, we can account for this term with weights $A_t$ emerging as the solution of a different differential equation coming from an adaptation of the Jarzynski equality (Jarzynski, 1997; Vaikuntanathan & Jarzynski, 2008):

**Proposition 2.1** (Jarzynski's estimator). *Assume that $r_0 = 0$ so that $\rho_0^r = \rho_0$. Let $\tilde{x}_t$ solve the SDE* (12) *with $\tilde{x}_0 \sim \rho_0^r$ and define*

$$A_t = \int_0^t \left( b_s(\tilde{x}_s) \cdot \nabla r_s(\tilde{x}_s) + \partial_s r_s(\tilde{x}_s) \right) ds, \tag{17}$$

*Then for all $t \in [0, 1]$ and any test function $h : \mathbb{R}^d \to \mathbb{R}$, we have*

$$\int_{\mathbb{R}^d} h(x)\hat{\rho}_t(x)dx = \frac{\mathbb{E}[e^{A_t}h(\tilde{x}_t)]}{\mathbb{E}[e^{A_t}]}, \tag{18}$$

*where the expectations at the right-hand side are taken over the law of $\tilde{x} = (\tilde{x}_t)_{t \in [0,T]}$.*

The proof of this statement in Appendix A.1 relies on manipulating the augmented FPE of the joint PDF $f_t(x, a)$ of $(\tilde{x}_t, A_t)$. This relation ensures that the lag associated to naively using the gradient of

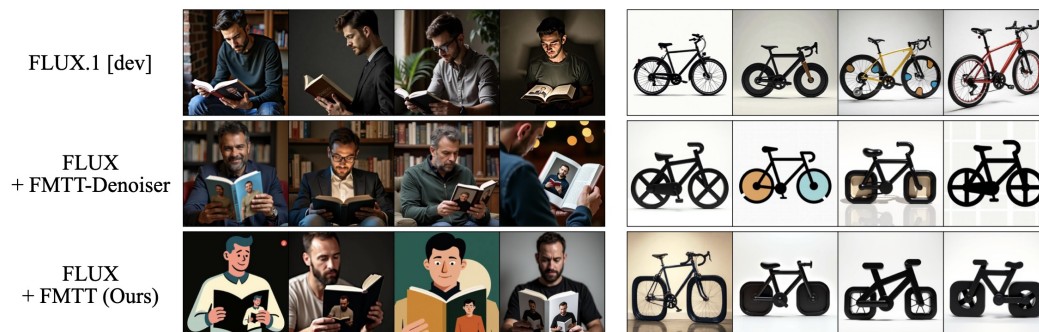

FLUX.1 [dev]

FLUX + FMTT-Denoiser

FLUX + FMTT (Ours)

*Prompt: "A man reading a book that shows a picture of the same man reading the same book"*      *Prompt: "A bicycle with **square wheels**"*

**Figure 3:** Qualitative results using VLM-based rewards. Prompts where the base model fails to generate aligned outputs are corrected by FMTT, with flow map look-ahead producing the most reliable improvements.

the reward in the diffusion can be compensated for by reweighting the trajectories, and, in addition, these weights can be used to perform resampling of the trajectories as is done in Sequential Monte Carlo (SMC) and birth/death processes, as is depicted in Figure 2. Here, as the trajectories walk out, the walkers can be *resampled* using the importance weights, removing some and duplicating others.

**Simplicity of importance weights with the flow map.** Interestingly, the importance weights in (17) take on a remarkably simple form when we use as $r_t(x)$ the reward composed with the flow map, as stated in the following proposition

> **Proposition 2.2** (Unbiased Flow Map Trajectory Tilting). *Using the same notations as in Proposition 2.1, if $r_t(x) = t\,r(X_{t,1}(x))$, then the importance weights defined (17) reduce to*
>
> $$A_t = \int_0^t r(X_{s,1}(\tilde{x}_s))ds. \tag{19}$$

This result is proven in Appendix A.2, and relies on a simple modification of the proof of Proposition 2.1 and the Eulerian equation (4). Thanks to the flow map, the complicated derivatives appearing in (17) reduce to simply compounding the reward over the look-ahead trajectory.

**Reward-modified drift.** A variant of Proposition 2.1 makes use of not only augmenting the score with the gradient of the reward with the look-ahead, but also the drift itself. Because $v_{t,t}$ is the velocity field of a stochastic interpolant and is related to the score via (6), we can replace the SDE given in (13) with

$$d\tilde{x}_t = [v_{t,t}(\tilde{x}_t) + \chi_t \nabla r_t(\tilde{x}_t)]\,dt + \epsilon_t\,[s_t(\tilde{x}_t) + \nabla r_t(\tilde{x}_t)]\,dt + \sqrt{2\epsilon_t}dW_t \qquad \tilde{x}_0 \sim \rho_0, \tag{20}$$

where $(\chi_t)_{t\in[0,1]}$ is arbitrary, and the log-weight ODE with

$$d\tilde{A}_t = \big(b_t(\tilde{x}_t)\cdot\nabla r_t(\tilde{x}_t) + \partial_t r_t(\tilde{x}_t) + \chi_t\big(\|\nabla r_t(\tilde{x}_t)\|^2 + \Delta r_t(\tilde{x}_t) + \langle\nabla r_t(\tilde{x}_t), s_t(\tilde{x}_t)\rangle\big)\big)\,dt \tag{21}$$

where $\tilde{A}_0 = 0$. It is proven in Appendix A.3 that these equations also provide an unbiased sampler of the tilted distribution by ensuring (18) holds for any choice of $(\chi_t)_{t\in[0,1]}$, and three choices besides the default $\chi_t \equiv 0$ are studied. This further augments the sampling process toward the tilt and will prove useful in the experiments below.

**Characterizing the effectiveness of the flow map trajectory tilting** As discussed above, the dynamics (13)-(19) and (35)-(21) are simulated via SMC, which involves a system of $N$ particles, a time discretization $(t_k)_{k=1}^K$ and a (random) number of resampling steps $R \leq K$. SMC algorithms naturally yield an unbiased estimate $\hat{Z}_{\text{SMC}}$ of the normalization constant $\mathbb{E}[e^{A_t}]$, and a low variance $\text{Var}[\hat{Z}_{\text{SMC}}]$ is a proxy for an efficient sampling schedule, as it signals that the empirical distribution of the $N$ particles is close to $\hat{\rho}_1$. However, $\text{Var}[\hat{Z}_{\text{SMC}}]$ often takes exponentially high values, making

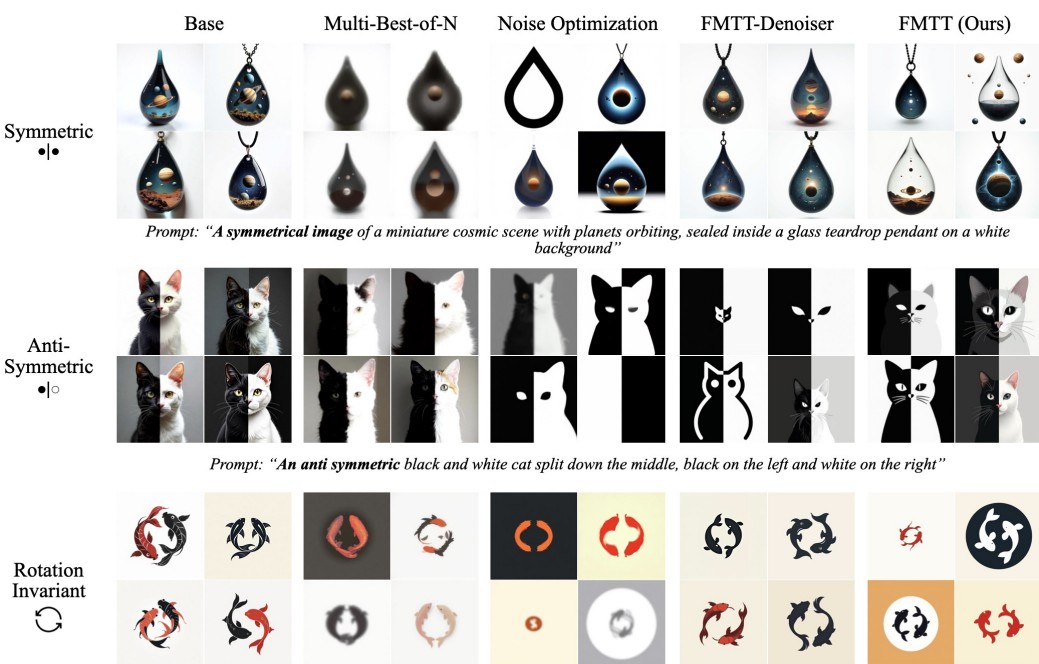

**Figure 4:** Qualitative comparison on three basic geometric rewards (symmetry, anti-symmetry, rotation invariance). The gradient-based methods that change the generative dynamics produce sharper images that satisfy the constraints more reliably than prior methods.

it hard to approximate, and it depends on $N$, $K$, and $R$. The following proposition introduces the **thermodynamic length**, a quantity related to $\text{Var}[\hat{Z}_{\text{SMC}}]$ which does not suffer from these issues.

> **Proposition 2.3** (Total discrepancy and thermodynamic length, informal)**.** *The variance* $\text{Var}[\hat{Z}_{\text{SMC}}]$ *can be expressed in terms of the number of particles $N$, discretization steps $K$, and resampling steps $R$, and the* total discrepancy $D(\mathcal{T})$ *which depends on discretization schedule* $\mathcal{T} = (t_k)_{k=0}^K$ *and is computable in practice. For optimal $\mathcal{T}$, $K\sqrt{D(\mathcal{T})}$ can be replaced by* thermodynamic length $\Lambda$*, which can be computed from the $\tilde{A}_t$ updates and that is agnostic to $\mathcal{T}$, $N$ and $R$, and satisfies that $\Lambda \leq K\sqrt{D(\mathcal{T})}$.*

**From sampling to search: making the most of rewards in practice.** Notably, the importance weights defined in either (19) or (21) do not need to be used to perform exact sampling. They can also be used to perform various greedy search algorithms that **search** for samples with high reward. That is, we are free to use top-n sampling approaches in place of the resampling one would usually do in SMC. Unlike (Ma et al., 2025), this use of top-n still makes use of the gradient of the reward along the trajectory, while also making use of better signal thanks to the flow map.

## 3 NUMERICAL EXPERIMENTS

Algorithm 1 details all the inference-time adaption techniques that we introduce. The base algorithm described in Proposition 2.2 corresponds to the choice $\epsilon_k = \epsilon_{t_k}$ and $\eta_k = 0$ for all $k = 0 : K$, and Sampling = True.

### 3.1 SAMPLING EXPERIMENTS ON MNIST

To demonstrate that the thermodynamic length is a meaningful diagnostic of the performance of the tilt, we compute the thermodynamic length on the problem of tilting an unconditional image generation model to a class conditional one. This will allow us to show that the process driven by

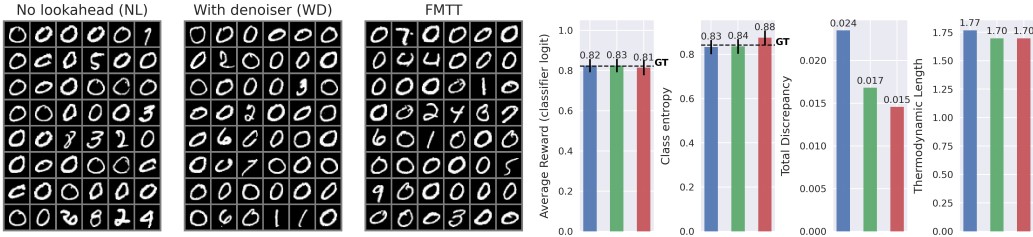

**Figure 5:** Comparison of MNIST tilted sampling to generate digits that would be classified as zeros. **Left**: using (12)-(17) with no look-ahead. **Center:** Doing the same with the denoiser composed with the reward. **Right**: Doing the same with the flow map i.e. our method FMTT. FMTT has the lowest total discrepancy, and the smallest thermodynamic length.

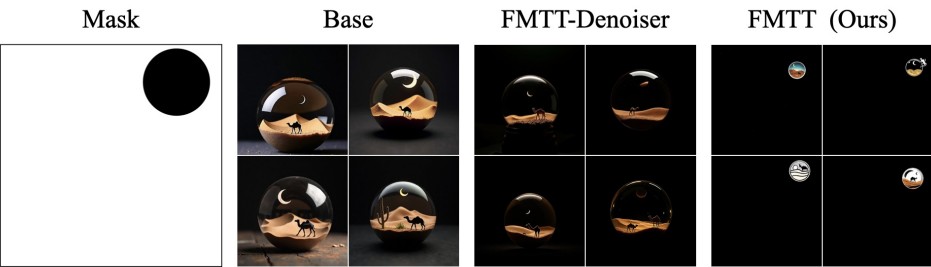

*Prompt: "A tiny desert landscape with sand dunes, a crescent moon above, and a lone camel silhouette, all inside a transparent glass orb **located in the top right of the image on a black background**"*

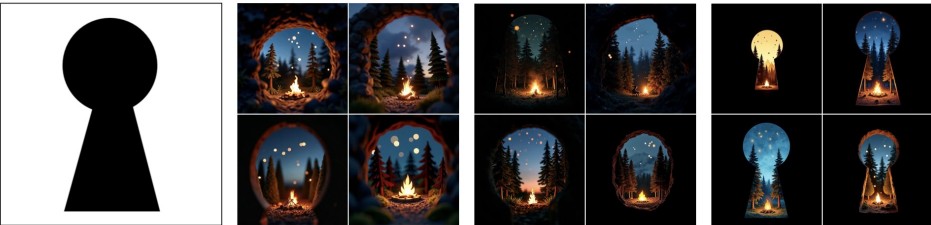

*Prompt: "A miniature forest with tall pine trees, a glowing campfire, and fireflies drifting in the night sky, **all inside a keyhole on a black background**"*

**Figure 6:** Qualitative comparison on masked rewards. Only our flow map-based FMTT reliably satisfies the constraints, concentrating content in the unmasked regions.

FMTT more efficiently samples the tilted distribution of interest. For sake of expediency to make the computation of the thermodynamic discrepancies and length calculable, we measure and compare these quantities on an MNIST experiment where the reward model is 0.1 times the likelihood that a classifier assigns to an unconditionally generated image being a zero. In Figure 7 we compare the three setups specified below equation (12) and we plot the average reward, which is the average log-likelihood of being a zero for the generated samples, the average class entropy assigned by the classifier. For the three algorithms, the ground truth values fall within the standard error bars, and the lowest total discrepancy and thermodynamic length correspond to FMTT. Appendix D contains a discussion of these results and additional experiments on MNIST.

## 3.2 TEXT-TO-IMAGE EXPERIMENTS

We evaluate our approach on text-to-image generation using the 4-step distilled flow map from Align Your Flow (Sabour et al., 2025), trained by distilling the open-source FLUX.1-dev model (Labs, 2024). We consider three categories of reward functions: 1) *Human preference rewards* capturing visual quality and text alignment, 2) *Geometric rewards* enforcing structural constraints such as symmetry or rotation invariance, 3) *VLM-based rewards* defined through natural language queries.

As baselines, we compare against *gradient-free* and *gradient-based* methods. Gradient-free approaches such as Best-of-$N$ (Chatterjee & Diaconis, 2018), Multi-Best-of-$N$ (Lee et al., 2025), and beam search (Fernandes et al., 2025) rely on sampling and selection, and remain confined to the

base model's distribution. Gradient-based methods use reward gradients, but differ in how they apply them: ReNO (Eyring et al., 2024) performs gradient ascent in the initial noise latent space, keeping samples tied to the base distribution, whereas our FMTT algorithm (and its ablations) use the gradient to modify the generative process itself, enabling exploration beyond the model's support and generation of out-of-distribution samples. Notably, the gradient of the reward used in FMTT efficiently gives meaningful signal for the whole trajectory thanks to the flow map, enably OOD sample generation for highly nuanced rewards.

**Human Preference Rewards.** To quantitatively benchmark FMTT, we follow prior work (Eyring et al., 2024) and use a linear combination of PickScore (Kirstain et al., 2023), HPSv2 (Wu et al., 2023b), ImageReward (Xu et al., 2023), and CLIPScore (Radford et al., 2021) as the reward and perform evaluation on GenEval (Ghosh et al., 2023), which consists of ≈550 object-centric prompts and measures the quality of generated images using a pre-trained object detector. Results in Table 1.

The base model, FLUX.1-dev, achieves strong scores due to its training on large amounts of object-centric data. Distillation into a 4-step flow map slightly reduces performance but significantly accelerates generation. A simple best-of-$N$ search on top of the flow map recovers this drop and surpasses the base diffusion model while remaining about 30% faster. More advanced search methods, such as multi-best-of-$N$ or beam search, yield additional but modest gains. Using reward gradients with FMTT provides a further small improvement. It is important to note that the base FLUX model has already been post-trained with human preference data. As a result, optimizing for the same types of reward during inference cannot substantially shift its output distribution, which explains why improvements across methods remain limited.

Finally, we ablate the use of the 4-step flow map look-ahead by comparing FMTT against variants using either a 1-step denoiser or a 4-step diffusion sampler. The flow map look-ahead consistently performs best, in line with our earlier findings.

**Geometric Transformation Rewards.** Recall that FLUX.1-dev has already been trained on human preference data, so its output distribution is already biased toward high preference rewards. This explains why much of the improvement in the previous experiments could be achieved with a simple best-of-$N$ search, with a slight additional boost being obtained when using gradient-based methods. This changes when the reward function is more specialized and achieves high values only in the long tails of the base model's output distribution. An example is a reward that enforces invariance under simple geometric transformations, defined as $r(x) = -d(x, T(x))$ where $T(x) : \mathbb{R}^d \to \mathbb{R}^d$ is a transformation function and $d(\cdot, \cdot)$ is a distance metric. For example, if $T$ is a masking operator, this reward incentivizes blackening the masked regions which can be used as a way to position elements in the scene. Similar rewards can be defined for symmetry, anti-symmetry, rotation, and so on.

Figure 4 shows that the base model roughly aligns with these objectives but does not fully satisfy them (the small planets break symmetry, the cats eyes aren't anti-symmetric, and the koi fish have different colors so is not rotation invariant). Prior methods such as multi-best-of-$N$ (Lee et al., 2025) and ReNO (Eyring et al., 2024) also fail, either breaking constraints or producing blurry images. In contrast, our gradient-based variants directly modify the dynamics, producing sharper outputs that more reliably satisfy the constraints. For a harder case, we evaluate the masked reward

**Table 1:** Quantitative results on GenEval.

| Method | Mean ↑ | Single Obj. ↑ | Two Obj. ↑ | Counting ↑ | Colors ↑ | Position ↑ | Attr. Binding ↑ | NFE |
|---|---|---|---|---|---|---|---|---|
| **Diffusions + Flow Maps** | | | | | | | | |
| FLUX.1 [dev] | 0.65 | 0.99 | 0.78 | 0.70 | 0.78 | 0.18 | 0.45 | 180 |
| Flow Map | 0.62 | 0.99 | 0.72 | 0.63 | 0.80 | 0.19 | 0.39 | 16 |
| **Gradient-Free Search** | | | | | | | | |
| FLUX.1 [dev] + Best-of-$N$ | 0.75 | 0.99 | 0.94 | 0.83 | 0.86 | 0.26 | 0.57 | 1440 |
| Flow Map + Best-of-$N$ | 0.73 | 1.00 | 0.88 | 0.82 | 0.85 | 0.25 | 0.59 | 128 |
| Flow Map + Multi-best-of-$N$ | 0.76 | 1.00 | 0.95 | 0.84 | 0.85 | 0.26 | 0.69 | 1280 |
| Flow Map + Beam Search | 0.75 | 1.00 | 0.92 | 0.86 | 0.85 | 0.29 | 0.58 | 1200 |
| **Gradient-Based Search** | | | | | | | | |
| Flow Map + ReNO | 0.71 | 0.98 | 0.89 | 0.79 | 0.89 | 0.20 | 0.57 | 1280 |
| FMTT (Ours) | **0.79** | **1.0** | **0.97** | **0.90** | **0.91** | **0.30** | **0.64** | 1400 |
| FMTT - 1-step denoiser lookahead | 0.75 | 0.99 | 0.90 | 0.87 | 0.87 | 0.26 | 0.59 | 350 |
| FMTT - 4-step diffusion lookahead | 0.75 | 0.99 | 0.93 | 0.86 | 0.89 | 0.27 | 0.57 | 1400 |

in Figure 6. FMTT with a denoiser look-ahead produces darker images with higher rewards than the base model, but fails to move all content to the unmasked region. Using a flow map look-ahead, however, successfully maximizes the reward, generating images that fully satisfy the constraint.

**VLMs as a Judge.**   We explore using pretrained VLMs to judge our images. The setup is straightforward: we provide the generated image along with a binary yes/no question, and define the reward as the difference between the log-probabilities of the answers "Yes" and "No". This formulation allows rewards to be expressed entirely in natural language. Since some VLMs accept multiple image inputs, we can also define rewards that depend on comparisons between the generated image and additional context images. In our experiments, we use Skywork-VL Reward (Wang et al., 2025a) for single-image settings and Qwen2.5-VL-7B-Instruct (Bai et al., 2025) for multi-image applications.

Qualitative results are shown in Figure 3. The figure highlights two prompts where the base model fails to produce text-aligned outputs. When the VLM is used as a reward to judge whether the prompt is a correct caption for the image, our FMTT algorithm generates outputs that match the input text much more accurately. While FMTT with a denoiser look-ahead improves text alignment in some cases, its success rate is low (only 1/4 images match the prompt). By contrast, FMTT with a flow map look-ahead consistently produces better-aligned images, such as correctly repeating characters in the book example, and does so with a higher success rate and average final reward. For additional VLM-based experiments, including multi-image settings, please see Appendix E.

The reward here is based on the question: *"Is {PROMPT} a correct caption for the image? Please answer no if the image is not in high definition (i.e., clear, sharp, not pixelated, and not blurry)."*

One caveat is the possibility of reward hacking (Amodei et al., 2016), as the search procedure explicitly maximizes the VLM reward. To mitigate this, the yes/no questions must be written with enough detail to prevent the algorithm from exploiting loopholes. Discussion and examples in Appendix F.

**Conclusions.**   We have presented FMTT, using flow maps for improved test-time scaling of diffusions. We envision that FMTT can overcome the limitations of common image generation systems when nuanced control is required or challenging rewards are given, for instance by VLMs.

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

# Appendix

## A  PROOFS

### A.1  PROOF OF PROPOSITION 2.1

Consider the coupled SDE/ODE:

$$d\tilde{x}_t = \big(b_t(\tilde{x}_t) + \epsilon_t s_t(\tilde{x}_t) + \epsilon_t \nabla r_t(\tilde{x}_t)\big)dt + \sqrt{2\epsilon_t}dW_t, \qquad \tilde{x}_0 \sim \rho_0,$$

$$dA_t = \big(b_t(\tilde{x}_t) \cdot \nabla r_t(\tilde{x}_t) + \partial_t r_t(\tilde{x}_t)\big)dt, \qquad A_0 = 0. \tag{22}$$

Let $f_t(x, a)$ be the joint probability density of $(\tilde{x}_t, A_t)$. Then, it satisfies the Fokker-Planck equation

$$\partial_t f_t(x, a) = -\nabla_x \cdot \big(\big(b_t(x) + \epsilon_t s_t(x) + \epsilon_t \nabla r_t(x)\big)f_t(x, a)\big) - \partial_a \big(\big(b_t(x) \cdot \nabla r_t(x) + \partial_t r_t(x)\big)f_t(x, a)\big)$$

$$+ \epsilon_t \Delta_x f_t(x, a),$$

$$f_t(x, a) = \rho_0(x)\delta_0(a). \tag{23}$$

Observe that if we let $\bar{\rho}_t(x) = \int_{\mathbb{R}} f_t(x, a)e^a \, da$, $\bar{\rho}$ satisfies

$$\partial_t \bar{\rho}_t(x) = \int_{\mathbb{R}} \partial_t f_t(x, a)e^a \, da$$

$$= \int_{\mathbb{R}} \Big( -\nabla_x \cdot \big(\big(b_t(x) + \epsilon_t s_t(x) + \epsilon_t \nabla r_t(x)\big)f_t(x, a)\big) - \big(b_t(x) \cdot \nabla r_t(x) + \partial_t r_t(x)\big)\partial_a f_t(x, a)$$

$$+ \epsilon_t \Delta_x f_t(x, a)\Big)e^a \, da$$

$$= -\nabla_x \cdot \Big(\big(b_t(x) + \epsilon_t s_t(x) + \epsilon_t \nabla r_t(x)\big)\int_{\mathbb{R}} f_t(x, a)e^a \, da\Big)$$

$$- \big(b_t(x) \cdot \nabla r_t(x) + \partial_t r_t(x)\big)\int_{\mathbb{R}} \partial_a f_t(x, a)e^a \, da + \epsilon_t \Delta_x \Big(\int_{\mathbb{R}} f_t(x, a)e^a \, da\Big)$$

$$= -\nabla_x \cdot \big(\big(b_t(x) + \epsilon_t s_t(x) + \epsilon_t \nabla r_t(x)\big)\bar{\rho}_t(x)\big) + \big(b_t(x) \cdot \nabla r_t(x) + \partial_t r_t(x)\big)\bar{\rho}_t(x) + \epsilon_t \Delta_x \bar{\rho}_t(x). \tag{24}$$

where the fourth inequality holds through integration by parts:

$$\int_{\mathbb{R}} \partial_a f_t(x, a)e^a \, da = [f_t(x, a)e^a]_{-\infty}^{\infty} - \int_{\mathbb{R}} f_t(x, a)\partial_a e^a \, da = -\int_{\mathbb{R}} f_t(x, a)e^a \, da = -\bar{\rho}_t(x). \tag{25}$$

We can reinterpret equation (24) as stating that for any test function $h$,

$$\int_{\mathbb{R}^d} h(x)\bar{\rho}_t(x) \, dx = \int_{\mathbb{R}^d} \int_{\mathbb{R}} f_t(x, a)e^a h(x) \, da \, dx = \mathbb{E}[e^{A_t}h(\tilde{x}_t)]. \tag{26}$$

However, $\bar{\rho}_t$ is not a normalized density for $t \geq 0$: if we integrate both sides of (24) over $\mathbb{R}^d$, and we use the divergence theorem, we obtain that

$$\partial_t \int_{\mathbb{R}^d} \bar{\rho}_t(x) \, dx = \int_{\mathbb{R}^d} \big(b_t(x) \cdot \nabla r_t(x) + \partial_t r_t(x)\big)\bar{\rho}_t(x) \, dx, \qquad \int_{\mathbb{R}^d} \bar{\rho}_0(x) \, dx = 1 \tag{27}$$

If we define $F_t = \int_{\mathbb{R}^d} \bar{\rho}_t(x) \, dx$, and we define $\hat{\rho}_t(x) = \bar{\rho}_t(x)/F_t$, we obtain that

$$\partial_t \hat{\rho}_t(x) = -\nabla_x \cdot \Big(\big(b_t(x) + \epsilon_t s_t(x) + \epsilon_t \nabla r_t(x)\big)\frac{\bar{\rho}_t(x)}{F_t}\Big) + \big(b_t(x) \cdot \nabla r_t(x) + \partial_t r_t(x)\big)\frac{\bar{\rho}_t(x)}{F_t} + \epsilon_t \Delta_x \frac{\bar{\rho}_t(x)}{F_t}$$

$$- \frac{\partial_t F_t}{F_t}\frac{\bar{\rho}_t(x)}{F_t}, \tag{28}$$

and by (27), if we define $\hat{F}_t = \log F_t$, we have that

$$\partial_t \hat{F}_t = \frac{\partial_t F_t}{F_t} = \int_{\mathbb{R}^d} \big(b_t(x) \cdot \nabla r_t(x) + \partial_t r_t(x)\big)\frac{\bar{\rho}_t(x)}{F_t} \, dx = \int_{\mathbb{R}^d} \big(b_t(x) \cdot \nabla r_t(x) + \partial_t r_t(x)\big)\hat{\rho}_t(x) \, dx. \tag{29}$$

Plugging this into the right-hand side of (28) and substituting $\hat{\rho}_t(x) = \bar{\rho}_t(x)/F_t$ yields a PDE for $\hat{\rho}_t$ which matches (16):

$$\partial_t \hat{\rho}_t(x) = -\nabla_x \cdot \left( \left( b_t(x) + \epsilon_t s_t(x) + \epsilon_t \nabla r_t(x) \right) \hat{\rho}_t(x) \right) + \left( b_t(x) \cdot \nabla r_t(x) + \partial_t r_t(x) - \partial_t \hat{F}_t \right) \hat{\rho}_t(x) + \epsilon_t \Delta_x \hat{\rho}_t(x).$$
(30)

To show that equation (18) holds, we rely on (26) and the fact that $F_t = \int_{\mathbb{R}^d} \int_{\mathbb{R}} f_t(x, a) e^a \, da \, dx = \mathbb{E}[e^{A_t}]$:

$$\int_{\mathbb{R}^d} h(x) \hat{\rho}_t(x) \, dx = \frac{1}{F_t} \int_{\mathbb{R}^d} h(x) \bar{\rho}_t(x) \, dx = \frac{\mathbb{E}[e^{A_t} h(\tilde{x}_t)]}{\mathbb{E}[e^{A_t}]}.$$
(31)

## A.2 Proof of Proposition 2.2

When $r_t(x) = tr(X_{t,1}(x))$, the log-weight $A_t$ defined in (17) satisfies the ODE

$$\begin{aligned}
\frac{\mathrm{d}A_t}{\mathrm{d}t} &= t \, b_t(\tilde{x}_t) \cdot \nabla_{\tilde{x}_t} r(X_{t,1}(\tilde{x}_t)) + \partial_t \left( t \, r(X_{t,1}(\tilde{x}_t)) \right) \\
&= t b_t(\tilde{x}_t) \cdot \nabla X_{t,1}(\tilde{x}_t)^\top \nabla r(X_{t,1}(\tilde{x}_t)) + r(X_{t,1}(\tilde{x}_t)) + t \, \partial_t X_{t,1}(\tilde{x}_t) \cdot \nabla r(X_{t,1}(\tilde{x}_t)) \\
&= t \nabla r(X_{t,1}(\tilde{x}_t)) \cdot \left( \partial_t X_{t,1}(\tilde{x}_t) + \nabla X_{t,1}(\tilde{x}_t) b_t(\tilde{x}_t) \right) + r(X_{t,1}(\tilde{x}_t)),
\end{aligned}$$
(32)

The Eulerian identity states that

$$\partial_t X_{t,1}(x) + \nabla X_{t,1}(x) b_t(x) = 0.$$
(33)

To prove (33), we write $0 = \partial_s (X_{s,1} \circ X_{t,s})(x) = \partial_s X_{s,1}(X_{t,s}(x)) + \nabla X_{s,1}(X_{t,s}(x)) \partial_s X_{t,s}(x)$, we use that $\partial_s X_{t,s}(x) = b(X_{t,s}(x))$, and we set $s = t$. Plugging (33) into the right-hand side of (32) yields

$$\frac{\mathrm{d}A_t}{\mathrm{d}t} = r(X_{t,1}(\tilde{x}_t)),$$
(34)

which concludes the proof.

## A.3 Modifying the drift with the flow map reward

While the position and log-weight dynamics in Proposition 2.1 and Proposition 2.2 are remarkably simple to compute, they are not the only ones that can be used to sample from the tilted distribution. In particular, if we add an additional term $\chi_t \nabla r_t(\tilde{x}_t)$ to the position SDE, we can adjust the log-weights dynamics so that the coupled system still lets us compute expectations according to the tilted distribution $\hat{\rho}_t$ for all $t \in [0, 1]$. Namely, the reward-modified position SDE reads:

$$d\tilde{x}_t = [v_{t,t}(\tilde{x}_t) + \chi_t \nabla r_t(\tilde{x}_t)] \, dt + \epsilon_t [s_t(\tilde{x}_t) + \nabla r_t(\tilde{x}_t)] \, dt + \sqrt{2\epsilon_t} dW_t \qquad \tilde{x}_0 \sim \rho_0, \quad (35)$$

There are four choices of $\chi_t$ of particular interest:

(i) **Default dynamics**: $\chi_t = 0$. This choice recovers the dynamics introduced in Section 2.3.

(ii) **Tilted score dynamics**: $\chi_t = \eta_t := \alpha_t(\frac{\dot{\beta}_t}{\beta_t} \alpha_t - \dot{\alpha}_t)$. When the dynamics (1) has been learned using stochastic interpolants, the score $s_t(x)$ and the vector field $b_t(x)$, or equivalently $v_{t,t}(x)$, are related to each other via the equation

$$b_t(x) = \alpha_t(\frac{\dot{\beta}_t}{\beta_t} \alpha_t - \dot{\alpha}_t) s_t(x) + \frac{\dot{\beta}_t}{\beta_t} x$$
(36)

which (6) is a particular case of, when $\alpha_t = 1 - t$, $\beta_t = t$. The score of the tilted distribution at time $t$ is $\hat{s}_t(x) = \nabla \log \hat{\rho}_t(x) = s_t(x) + \nabla r_t(x)$, and if we replace $s_t(x)$ by $\hat{s}_t(x)$ in (36), we obtain

$$b_t(x) = \alpha_t(\frac{\dot{\beta}_t}{\beta_t} \alpha_t - \dot{\alpha}_t) \hat{s}_t(x) + \frac{\dot{\beta}_t}{\beta_t} x = v_{t,t}(x) + \alpha_t(\frac{\dot{\beta}_t}{\beta_t} \alpha_t - \dot{\alpha}_t) \nabla r_t(x).$$
(37)

(iii) **Local tilt dynamics**: $\chi_t = \epsilon_t$. If we let $P(x_{(k+1)h}|x_{kh})$ be the transition kernel for the Euler-Maruyama discretization of the SDE

$$d\tilde{x}_t = \left(v_{t,t}(\tilde{x}_t) + \epsilon_t s_t(\tilde{x}_t) + \epsilon_t \nabla r_t(\tilde{x}_t)\right)dt + \sqrt{2\epsilon_t}dW_t, \tag{38}$$

the local tilt dynamics arises from tilting $P(x_{(k+1)h}|x_{kh})$ according to the reward $r_{(k+1)h}(x_{(k+1)h})$, which yields the transition kernel $P^r(x_{(k+1)h}|x_{kh}) \propto P(x_{(k+1)h}|x_{kh})\exp(r_{(k+1)h}(x_{(k+1)h}))$. By Lemma A.4, when $h \to 0$, $P^r(x_{(k+1)h}|x_{kh})$ can be viewed as the transition kernel for the SDE

$$d\tilde{x}_t = [v_{t,t}(\tilde{x}_t) + \epsilon_t \nabla r_t(\tilde{x}_t)]\,dt + \epsilon_t\,[s_t(\tilde{x}_t) + \nabla r_t(\tilde{x}_t)]\,dt + \sqrt{2\epsilon_t}dW_t, \tag{39}$$

which amounts to setting $\chi_t = \epsilon_t$ in (35).

(iv) **Base dynamics**: $\chi_t = -\epsilon_t$. With this choice, the SDE (35) becomes

$$d\tilde{x}_t = [v_{t,t}(\tilde{x}_t) + \epsilon_t s_t(\tilde{x}_t)]\,dt + \sqrt{2\epsilon_t}dW_t, \tag{40}$$

which does not involve the reward $r_t$.

The following proposition, which generalizes Proposition 2.1, shows the log-weight dynamics that we must use for generic reward multipliers $\chi_t$, written in three different ways. Observe that

---

**Proposition A.1** (Unbiased Map Tilting with reward-modified vector field). *Let $(\chi_t)_{t\in[0,1]}$ be an arbitrary real valued function, and let $r_t$ be an arbitrary time-dependent reward. Let $(\tilde{x}_t, A_t)$ be a solution of*

$$d\tilde{x}_t = \left(b_t(\tilde{x}_t) + \chi_t \nabla r_t(\tilde{x}_t) + \epsilon_t s_t(\tilde{x}_t) + \epsilon_t \nabla r_t(\tilde{x}_t)\right)dt + \sqrt{2\epsilon_t}dW_t, \qquad \tilde{x}_0 \sim \rho_0, \tag{41}$$

$$dA_t = \left(b_t(\tilde{x}_t)\cdot\nabla r_t(\tilde{x}_t) + \partial_t r_t(\tilde{x}_t) + \chi_t\left(\|\nabla r_t(\tilde{x}_t)\|^2 + \Delta r_t(\tilde{x}_t) + \langle\nabla r_t(\tilde{x}_t), s_t(\tilde{x}_t)\rangle\right)\right)dt, \ A_0 = 0. \tag{42}$$

*Then for all $t \in [0,1]$ and any test function $h : \mathbb{R}^d \to \mathbb{R}$, we have*

$$\int_{\mathbb{R}^d} h(x)\hat{\rho}_t(x)dx = \frac{\mathbb{E}[e^{A_t}h(\tilde{x}_t)]}{\mathbb{E}[e^{A_t}]}, \tag{43}$$

*where the expectations at the right-hand side are taken over the law of $(\tilde{x}, A)$. The equation (42) for $A_t$ can be rewritten as*

$$dA_t = \left(b_t(\tilde{x}_t)\cdot\nabla r_t(\tilde{x}_t) + \partial_t r_t(\tilde{x}_t) + \chi_t\left(\|\nabla r_t(\tilde{x}_t)\|^2 + \langle\nabla r_t(\tilde{x}_t), s_t(\tilde{x}_t)\rangle\right)\right)dt$$
$$+ \frac{\chi_t}{\sqrt{2\epsilon_t}}\nabla r_t(\tilde{x}_t)\cdot\mathrm{d}^- W_t - \frac{\chi_t}{\sqrt{2\epsilon_t}}\nabla r_t(\tilde{x}_t)\cdot dW_t, \ A_0 = 0. \tag{44}$$

*Equation (42) can be rewritten in a third way as*

$$\mathrm{d}A_t = \begin{cases} \frac{\mathrm{d}}{\mathrm{d}s}\mathbb{E}[e^{r_s(X_s^\chi) - r_t(\tilde{x}_t)} \mid X_t^\chi = \tilde{x}_t]\big|_{s=t}\,\mathrm{d}t, & \text{if } \chi_t \geq 0, \\ \left(2\frac{\mathrm{d}}{\mathrm{d}s}r_s(X_s^0)\big|_{\substack{X_t^0=\tilde{x}_t \\ s=t}} - \frac{\mathrm{d}}{\mathrm{d}s}\mathbb{E}[e^{r_s(X_s^\chi) - r_t(\tilde{x}_t)} \mid X_t^\chi = \tilde{x}_t]\big|_{s=t}\right)\mathrm{d}t, & \text{if } \chi_t < 0, \end{cases} \tag{45}$$
$$A_0 = 0,$$

*where $X^\chi$ is a solution of the SDE*

$$\mathrm{d}X_t^\chi = \left(b_t(X_t^\chi) + |\chi_t|s_t(X_t^\chi)\right)\mathrm{d}t + \sqrt{2|\chi_t|}dW_t, \tag{46}$$

*and $X^0$ is a solution of the ODE $\mathrm{d}X_t^0 = b_t(X_t^\chi)\,\mathrm{d}t$.*

---

*Proof.* **(i) Proof of equation** (43). When we replace $b_t(x)$ by $\tilde{b}_t(x) = b_t(x) + \chi_t\nabla r_t(x)$, the analog of equation (15) is:

$$\partial_t\hat{\rho}_t + \nabla\cdot(\tilde{b}_t\hat{\rho}_t) = \partial_t(e^{r_t+\hat{F}_t}\rho_t) + \nabla\cdot(\tilde{b}_t e^{r_t+\hat{F}_t}\rho_t)$$

$$= (\partial_t r_t + \partial_t\hat{F}_t)\hat{\rho}_t + e^{r_t+\hat{F}_t}\partial_t\rho_t + (b_t + \chi_t\nabla r_t)\cdot\nabla r_t\hat{\rho}_t + e^{r_t+\hat{F}_t}\nabla\cdot((b_t + \chi_t\nabla r_t)\rho_t)$$

$$= (\partial_t r_t + \partial_t\hat{F}_t)\hat{\rho}_t + b_t\cdot\nabla r_t\hat{\rho}_t + \chi_t\|\nabla r_t\|^2\hat{\rho}_t + \chi_t e^{r_t+\hat{F}_t}\nabla\cdot(\nabla r_t\rho_t)$$

$$= \left(b_t\cdot\nabla r_t + \partial_t r_t + \chi_t\left(\|\nabla r_t\|^2 + \Delta r_t + \langle\nabla r_t, s_t\rangle\right) + \partial_t\hat{F}_t\right)\hat{\rho}_t, \tag{47}$$

where the third equality holds because $\partial_t \rho_t + \nabla \cdot (b_t \rho_t) = 0$ by the FPE (8) with $\epsilon_t \equiv 0$, and in the fourth equality we used the definition $s_t(x) = \nabla \log \rho_t(x)$. Hence, when we replace $b_t$ by $\tilde{b}_t$, the terms $b_t \cdot \nabla r_t + \partial_t r_t$ get replaced by $b_t \cdot \nabla r_t + \partial_t r_t + \chi_t(\|\nabla r_t\|^2 + \Delta r_t + \langle \nabla r_t, s_t \rangle)$.

Given equation (47), the proof of the statement (43) is analogous to the proof of Proposition 2.1 in Appendix A.1, simply replacing $b_t$ by $b_t + \chi_t \nabla r_t$, and $b_t \cdot \nabla r_t + \partial_t r_t$ by $b_t \cdot \nabla r_t + \partial_t r_t + \chi_t(\|\nabla r_t\|^2 + \Delta r_t + \langle \nabla r_t, s_t \rangle)$. We omit a full proof.

**(ii) Proof of equation** (44). We proceed to prove that (44) is equivalent to (42). Observe that the solution of equation (42) is

$$A_t = \int_0^t \left( b_t(\tilde{x}_t) \cdot \nabla r_t(\tilde{x}_t) + \partial_t r_t(\tilde{x}_t) + \chi_\tau(\|\nabla r_\tau(\tilde{x}_\tau)\|^2 + \Delta r_\tau(\tilde{x}_\tau) + \langle \nabla r_\tau(\tilde{x}_\tau), s_\tau(\tilde{x}_\tau)\rangle) \right) dt. \tag{48}$$

Next, we handle the term $\nabla r_t$. Applying Lemma A.5, we obtain that

$$\int_0^t \chi_\tau \Delta r_\tau(\tilde{x}_\tau) \, d\tau = \int_0^t \tau \chi_\tau \Delta(r \circ X_{\tau,1})(\tilde{x}_\tau) \, d\tau$$
$$= \int_t^{t'} \frac{\tau \chi_\tau}{\sqrt{2\epsilon_\tau}} \nabla(r \circ X_{\tau,1})(\tilde{x}_\tau) \cdot d^- W_\tau - \int_t^{t'} \frac{\tau \chi_\tau}{\sqrt{2\epsilon_\tau}} \nabla(r \circ X_{\tau,1})(\tilde{x}_\tau) \cdot dW_\tau, \tag{49}$$

And plugging this back into (48) yields equation (44).

**(iii) Proof of equation** (45). Finally, we prove that (45) is equivalent to (42). We introduce the necessary concepts first. The semi-group for a (time-inhomogeneous) Markov process is defined as
$$P_{s,t} f(x) = \mathbb{E}[f(X_t) \,|\, X_s = x], \qquad 0 \leq s \leq t.$$
The (time-dependent) infinitesimal generator is defined by
$$L_t f_t(x) = \lim_{h \downarrow 0} \frac{P_{t,t+h} f_{t+h}(x) - f_t(x)}{h} = \frac{d}{ds} \mathbb{E}[f_s(X_s) \,|\, X_t = x]\big|_{s=t}. \tag{50}$$
For the process $X^\chi$ that solves (46), the infinitesimal generator takes the form
$$L_t^\chi f_t(x) = \partial_t f_t(x) + (b_t(x) + |\chi_t| s_t(x)) \cdot \nabla f_t(x) + |\chi_t| \Delta f_t(x). \tag{51}$$
We define the Hamilton-Jacobi-Bellman (HJB) residual $\mathcal{R}_t^\chi$ for the reward $r_t$ and the process $X^\chi$ as
$$\mathcal{R}_t^\chi(x) = b_t(x) \cdot \nabla r_t(x) + \partial_t r_t(x) + |\chi_t|(\|\nabla r_t(x)\|^2 + \Delta r_t(x) + \nabla r_t(x) \cdot s_t(x)). \tag{52}$$
The term "HJB residual" stems from the fact that the Hamilton-Jacobi-Bellman equation for a time-dependent reward $r_t$ and process $X^\chi$ can be expressed as $\mathcal{R}_t^\chi(x) = 0$ for all $t$ and $x^*$.

We start with the case $\chi_t \geq 0$. The term in the right-hand side of (42) can be rewritten as follows:
$$b_t(x) \cdot \nabla r_t(x) + \partial_t r_t(x) + \chi_t(\|\nabla r_t(x)\|^2 + \Delta r_t(x) + \langle \nabla r_t(x), s_t(x) \rangle) = \mathcal{R}_t^\chi(x)$$
$$= e^{-r_t(x)}[L_t^\chi e^{r_t}](x) = e^{-r_t(x)} \frac{d}{ds} \mathbb{E}[e^{r_s(X_s^\chi)} \,|\, X_t^\chi = x]\big|_{s=t} = \frac{d}{ds} \mathbb{E}[e^{r_s(X_s^\chi) - r_t(x)} \,|\, X_t^\chi = x]\big|_{s=t}. \tag{53}$$

where the second equality holds by Lemma A.6, and the third equality holds by the definition of the infinitesimal generator in (50). This concludes the case $\chi_t \geq 0$.

We move on to the case $\chi_t \geq 0$. The term in the right-hand side of (42) can be rewritten as follows:
$$b_t(x) \cdot \nabla r_t(x) + \partial_t r_t(x) + \chi_t(\|\nabla r_t(x)\|^2 + \Delta r_t(x) + \langle \nabla r_t(x), s_t(x) \rangle)$$
$$= 2(b_t(x) \cdot \nabla r_t(x) + \partial_t r_t(x))$$
$$\quad - (b_t(x) \cdot \nabla r_t(x) + \partial_t r_t(x) + \chi_t(\|\nabla r_t(x)\|^2 + \Delta r_t(x) + \langle \nabla r_t(x), s_t(x) \rangle))$$
$$= 2\mathcal{R}_t^0(x) - \mathcal{R}_t^\chi(x) = 2\frac{d}{ds}(e^{r_s(X_s^0) - r_t(x)})\big|_{\substack{X_t^0 = x \\ s=t}} - \frac{d}{ds} \mathbb{E}[e^{r_s(X_s^\chi) - r_t(\tilde{x}_t)} \,|\, X_t^\chi = x]\big|_{s=t} \tag{54}$$
$$= 2\frac{d}{ds} r_s(X_s^0)\big|_{\substack{X_t^0 = x \\ s=t}} - \frac{d}{ds} \mathbb{E}[e^{r_s(X_s^\chi) - r_t(\tilde{x}_t)} \,|\, X_t^\chi = x]\big|_{s=t}$$

where $\mathcal{R}_t^0$ is defined as $\mathcal{R}_t^\chi$ with $\chi = 0$, and the last equality holds by equation (53) applied to $\mathcal{R}_t^0$ and $\mathcal{R}_t^\chi$. $\qquad \square$

---

*Note that in stochastic optimal control, the HJB equation is usually written for the value function $V_t = -r_t$.

The following corollary particularizes Proposition A.1 to the case in which the time-dependent reward is built from the flow map $X_{t,1}$, thus generalizing Proposition 2.2.

**Corollary A.2** (Unbiased Map Tilting with reward-modified vector field from flow map). *When* $r_t(x) = tr(X_{t,1}(x))$, *equations* (42), (44) *and* (45) *simplify respectively to*

$$\mathrm{d}A_t = \big(\frac{r_t(\tilde{x}_t)}{t} + \chi_t\big(\|\nabla r_t(\tilde{x}_t)\|^2 + \Delta r_t(\tilde{x}_t) + \langle\nabla r_t(\tilde{x}_t), s_t(\tilde{x}_t)\rangle\big)\big)\,\mathrm{d}t, \ A_0 = 0. \tag{55}$$

$$\begin{aligned}\mathrm{d}A_t = &\big(\frac{r_t(\tilde{x}_t)}{t} + \chi_t\big(\|\nabla r_t(\tilde{x}_t)\|^2 + \langle\nabla r_t(\tilde{x}_t), s_t(\tilde{x}_t)\rangle\big)\big)\,\mathrm{d}t \\ &+ \frac{\chi_t}{\sqrt{2\epsilon_t}}\nabla r_t(\tilde{x}_t)\cdot\mathrm{d}^- W_t - \frac{\chi_t}{\sqrt{2\epsilon_t}}\nabla r_t(\tilde{x}_t)\cdot\mathrm{d}W_t, \ A_0 = 0.\end{aligned} \tag{56}$$

$$\mathrm{d}A_t = \begin{cases} \partial_s\mathbb{E}[e^{r_s(X_s^\chi) - r_t(\tilde{x}_t)}\,|\,X_t^\chi = \tilde{x}_t]\big|_{s=t}\,\mathrm{d}t, & \text{if } \chi_t \geq 0, \\ \big(\frac{2r_t(\tilde{x}_t)}{t} - \partial_s\mathbb{E}[e^{r_s(X_s^\chi) - r_t(\tilde{x}_t)}\,|\,X_t^\chi = \tilde{x}_t]\big|_{s=t}\big)\,\mathrm{d}t, & \text{if } \chi_t < 0, \end{cases} \quad A_0 = 0, \tag{57}$$

*Proof.* Using the argument in Appendix A.2 yields

$$b_t(x)\cdot\nabla r_t(x) + \partial_t r_t(x) = r(X_{t,1}(x)) = \frac{r_t(x)}{t}. \tag{58}$$

Substituting this into equations (42), (44) and (45) yields the simplified forms. □

**Remark A.3** (Setting $\chi_t = 0$ in Proposition A.1 and Corollary A.2). *Proposition A.1 and Corollary A.2 show that the default choice $\chi_t = 0$ is the only one for which the evolution of $A_t$ simplifies. Namely, when $\chi_t = 0$, equations* (42), (44) *and* (45) *in Proposition A.1 simplify respectively to*

$$\mathrm{d}A_t = \big(b_t(\tilde{x}_t)\cdot\nabla r_t(\tilde{x}_t) + \partial_t r_t(\tilde{x}_t)\big)\,\mathrm{d}t, \tag{59}$$

$$\mathrm{d}A_t = \big(b_t(\tilde{x}_t)\cdot\nabla r_t(\tilde{x}_t) + \partial_t r_t(\tilde{x}_t)\big)\,\mathrm{d}t, \tag{60}$$

$$\mathrm{d}A_t = \frac{\mathrm{d}}{\mathrm{d}s}r_s(X_s^0)\big|_{\substack{X_t^0 = \tilde{x}_t \\ s=t}}\,\mathrm{d}t, \tag{61}$$

*and equations* (55), (56), *and* (57) *in Corollary A.2 become*

$$\mathrm{d}A_t = \frac{r_t(\tilde{x}_t)}{t}\,\mathrm{d}t, \tag{62}$$

$$\mathrm{d}A_t = \frac{r_t(\tilde{x}_t)}{t}\,\mathrm{d}t, \tag{63}$$

$$\mathrm{d}A_t = \frac{\mathrm{d}}{\mathrm{d}s}r_s(X_s^0)\big|_{\substack{X_t^0 = \tilde{x}_t \\ s=t}}\,\mathrm{d}t, \tag{64}$$

### A.4 SOLVING THE SDE/ODE SYSTEM NUMERICALLY

We need to simulate the coupled dynamics on weights and positions numerically. The SDE (41) for $\tilde{x}_t$ can be solved via a $K$-step Euler-Maruyama scheme with time discretization $(t_k)_{k=0}^K$ and updates of the form

$$\begin{aligned}X_k = X_{k-1} + (t_k - t_{k-1})[&b_{t_{k-1}}(X_{k-1}) + \chi_{t_{k-1}}\nabla r_{t_{k-1}}(X_{k-1}) \\ &+ \epsilon_{t_{k-1}}(s_{t_{k-1}}(X_{k-1}) + \nabla r_{t_{k-1}}(X_{k-1}))] + \sqrt{2\epsilon_k(t_k - t_{k-1})}\xi_{k-1},\end{aligned} \tag{65}$$

where $\xi_{k-1} \sim N(0, \mathrm{I})$. For generic $\chi_t$, solving the dynamics for the log-weight $A_t$ requires one of the following approaches, which correspond to equations (42), (44) and (45), respectively:

(i) **Estimating the Laplacian** $\Delta r_t$: If solving (42), we need to estimate the Laplacian $\Delta r_t(x)$, which can be done using the Hutchinson trace estimator $\widehat{\Delta r_t}(x) = \frac{1}{2M\varepsilon}\sum_{m=1}^M z_m\cdot\big[\nabla r_t(x + $

$\varepsilon z_m) - \nabla r_t(x - \varepsilon z_m)\big]$, where $z_m \sim \mathcal{N}(0, I)$ or Rademacher, $\varepsilon \ll 1$. Then, the log-weights can be simulated with Euler updates of the form:

$$
\begin{aligned}
A_k = A_{k-1} + (t_k - t_{k-1}) \Big[ & b_{t_{k-1}}(X_{k-1}) \cdot \nabla r_{t_{k-1}}(X_{k-1}) + \partial_t r_{t_{k-1}}(X_{k-1}) \\
& + \chi_{t_{k-1}} \big( \|\nabla r_{t_{k-1}}(X_{k-1})\|^2 + \widehat{\Delta r_{t_{k-1}}}(X_{k-1}) \\
& + \langle \nabla r_{t_{k-1}}(X_{k-1}), s_{t_{k-1}}(X_{k-1}) \rangle \big) \Big]
\end{aligned}
\tag{66}
$$

When $\chi_{(k-1)h} = 0$, we do not need to compute the estimate the Laplacian, which makes the update much faster.

(ii) **Estimating the difference of forward and backward Itô integrals**: If solving (44), we estimate

$$
\int_{t_{k-1}}^{t_k} \frac{\chi_\tau}{\sqrt{2\epsilon_\tau}} \nabla r_\tau(X_{k-1}) \cdot \mathrm{d}^- W_\tau - \int_{t_{k-1}}^{t_k} \frac{\chi_\tau}{\sqrt{2\epsilon_\tau}} \nabla r_\tau(\tilde{x}_\tau) \cdot \mathrm{d}W_\tau
$$

$$
\approx \chi_{t_k} \sqrt{\frac{t_k - t_{k-1}}{2\epsilon_{t_k}}} \nabla r_{t_k}(X_k) \cdot \xi_{k-1} - \chi_{t_{k-1}} \sqrt{\frac{t_k - t_{k-1}}{2\epsilon_{t_{k-1}}}} \nabla r_{t_{k-1}}(X_{k-1}) \cdot \xi_{k-1}.
\tag{67}
$$

Thus, the update reads

$$
\begin{aligned}
A_k = A_{k-1} + (t_k - t_{k-1}) \Big[ & b_{t_{k-1}}(X_{k-1}) \cdot \nabla r_{t_{k-1}}(X_{k-1}) + \partial_t r_{t_{k-1}}(X_{k-1}) \\
& + \chi_{t_{k-1}} \big( \|\nabla r_{t_{k-1}}\|^2 + \nabla r_{t_{k-1}} \cdot s_{t_{k-1}} \big)(X_{k-1}) \Big] \\
& + \chi_{t_k} \sqrt{\frac{t_k - t_{k-1}}{2\epsilon_{t_k}}} \nabla r_{t_k}(X_{k-1}) \cdot \xi_{k-1}^n - \chi_{t_{k-1}} \sqrt{\frac{t_k - t_{k-1}}{2\epsilon_{t_{k-1}}}} \nabla r_{t_{k-1}}(X_{k-1}) \cdot \xi_{k-1}^n.
\end{aligned}
\tag{68}
$$

(iii) **Estimating expectations of** $\exp(r_t)$: If solving (45), we approximate

$$
\frac{\mathrm{d}}{\mathrm{d}s} \mathbb{E}[e^{r_s(X_s^\chi) - r_t(x)} \mid X_t^\chi = x]\big|_{s=t}
$$

$$
\begin{aligned}
\approx \frac{1}{M(t_k - t_{k-1})} \sum_{m=1}^M \exp \Big( & r_{t_k}(x + (t_k - t_{k-1})(b_{t_{k-1}}(x) + |\chi_{t_{k-1}}|s_{t_{k-1}}(x)) \\
& + \sqrt{2|\chi_{t_{k-1}}|(t_k - t_{k-1})} \xi_{k-1}^{(m)}) - r_{t_{k-1}}(x) \Big), \quad \xi_{k-1}^{(m)} \sim N(0, \mathrm{I}).
\end{aligned}
\tag{69}
$$

where we used a finite-difference approximation of the derivative, and the Euler-Maruyama update corresponding to the SDE (46). Thus, for $\chi_{(k-1)h} \geq 0$ the update reads

$$
\begin{aligned}
A_k = A_{k-1} + \frac{1}{M} \sum_{m=1}^M \exp \Big( & r_{t_k}(X_{k-1} + (t_k - t_{k-1})(b_{t_{k-1}}(X_{k-1}) + |\chi_{t_{k-1}}|s_{t_{k-1}}(X_{k-1})) \\
& + \sqrt{2|\chi_{t_{k-1}}|(t_k - t_{k-1})} \xi_{k-1}^{(m)}) - r_{t_{k-1}}(X_{k-1}) \Big), \quad \xi_{k-1}^{(m)} \sim N(0, \mathrm{I}).
\end{aligned}
\tag{70}
$$

Similarly,

$$
\frac{\mathrm{d}}{\mathrm{d}s} r_s(X_s^0)\big|_{\substack{X_t^0 = x \\ s=t}} \approx \frac{r_{t_k}(x + (t_k - t_{k-1})b_{t_k}(x)) - r_{t_k}(x)}{t_k - t_{k-1}}.
\tag{71}
$$

Thus, for $\chi_{(k-1)h} \leq 0$ the update reads

$$
\begin{aligned}
A_k = A_{k-1} + 2 \big( & r_{t_k}(X_{k-1} + (t_k - t_{k-1})b_{t_k}(X_{k-1})) - r_{t_{k-1}}(X_{k-1}) \big) \\
& - \frac{1}{M} \sum_{m=1}^M \exp \Big( r_{t_k}(X_{k-1} + (t_k - t_{k-1})(b_{t_{k-1}}(X_{k-1}) + |\chi_{t_{k-1}}|s_{t_{k-1}}(X_{k-1})) \\
& + \sqrt{2\epsilon_k(t_k - t_{k-1})} \xi_{k-1}^n) - r_{t_{k-1}}(X_{k-1}) \Big), \quad \xi_{k-1}^{(m)} \sim N(0, \mathrm{I}).
\end{aligned}
\tag{72}
$$

Observe that for $\chi_{(k-1)h} = 0$, equations (72) and (70) simplify to

$$A_k = A_{k-1} + r_{t_k}(X_{k-1} + (t_k - t_{k-1})b_{t_k}(X_{k-1})) - r_{t_{k-1}}(X_{k-1}), \tag{73}$$

which makes the update much less expensive.

Observe that following Corollary A.2, when $r_t(x) = tr(X_{t,1}(x))$ the terms $b_{t_{k-1}}(X_{k-1}) \cdot \nabla r_{t_{k-1}}(X_{k-1}) + \partial_t r_{t_{k-1}}(X_{k-1})$ in (66) and (68) can be replaced by $\frac{r_{t_{k-1}}(X_{k-1})}{t_{k-1}}$.

### A.4.1 On the best-performing approach in practice

When $\chi_t \neq 0$, the approach *(i)* requires evaluating the gradient $\nabla r_t$ at $M$ points to obtain an estimate with variance $\Theta(1/M)$, and the approach *(iii)* requires evaluating $r_t$ at $M$ points to obtain an estimate with variance $\Theta(1/M)$. Meanwhile, the approach *(ii)* only requires evaluating the gradient $\nabla r_t$ at the iterates $(X_k)_{k=0}^K$. Hence, a priori, the approach *(ii)* is preferable.

However, when the reward functions $r$ is a neural network, such as a classifier, a vision-language model, or a CLIP model, the gradients $\nabla r$ are very noisy as $r$ is highly non-smooth: if we perturb a sample $x$ with an amount of Gaussian noise small enough that the noiseless sample $x$ and the noised sample $\hat{x}$ have very similar reward values, and compute cosine similarity of the gradients $\nabla r(x)$ and $\nabla r(\hat{x})$, we generally obtain low values ($\ll 0.5$). The noisy gradients cause the difference of integrals in (67) to be poor, as the gradients $\nabla r_{t_k}(X_k)$ and $\nabla r_{t_{k-1}}(X_{k-1})$ are much more misaligned than they would be if the function was smoother. The noisy gradient issue also affect approach *(i)*, but it does not affect approach *(ii)*, which only relies on evaluations of $r_t$ but not its gradient. The issue can be mitigated by replacing the gradient evaluations $\nabla r_t$ by averages of the evaluations on noised samples, effectively estimating the gradient of $r_t$ convolved with a Gaussian. However, running approach *(ii)* with this mitigation requires using several evaluations of $\nabla r_t$ anyway. In practice, we observe that approach *(iii)* performs best, and that is the approach we used in our MNIST experiments in Section 3 and in Appendix D.

### A.5 Lemmas used in Appendix A.3

**Lemma A.4** (Simulating the local tilt dynamics). *Consider the Euler-Maruyama discretization of the SDE $d\tilde{x}_t = b_t^\epsilon(\tilde{x}_t)dt + \sqrt{2\epsilon_t}dW_t$, where $b_t^\epsilon(x) = b_t(x) + \epsilon_t s_t(x) + \epsilon_t \nabla r_t(x)$, which corresponds to the conditional density*

$$P(x_{(k+1)h}|x_{kh}) = \frac{\exp\left(-\|x_{(k+1)h} - x_{kh} - hb_{kh}^\epsilon(x_{kh})\|^2/(2h\epsilon_{kh})\right)}{(4\pi h\epsilon_{kh})^{d/2}}. \tag{74}$$

*If we define the conditional density*

$$P^r(x_{(k+1)h}|x_{kh}) \propto P(x_{(k+1)h}|x_{kh})\exp(r_{(k+1)h}(x_{(k+1)h})), \tag{75}$$

*in the limit $h \to 0$, this conditional density can be written as*

$$P^r(x_{(k+1)h}|x_{kh}) = \frac{\exp\left(-\|x_{(k+1)h} - x_{kh} - h(b_{kh}^\epsilon(x_{kh}) + \epsilon_{kh}\nabla r_{kh}(x_{kh}))\|^2/(2h\epsilon_{kh})\right)}{(4\pi h\epsilon_{kh})^{d/2}}, \tag{76}$$

*which is the conditional density for the Euler-Maruyama discretization of the SDE*

$$dX_t = (b_t^\epsilon(X_t) + \epsilon_t \nabla r_t(X_t))dt + \sqrt{2\epsilon_t}dB_t. \tag{77}$$

*Proof.* We use a discrete version of Itô's lemma:

$$r_{(k+1)h}(x_{(k+1)h}) \approx r_{kh}(x_{kh}) + h\left(\partial_t r_{kh}(x_{kh}) + \epsilon_{kh}\Delta r_{kh}(x_k)\right)$$
$$+ \langle \nabla r_{kh}(x_k), x_{(k+1)h} - x_{kh}\rangle + O(h^{3/2}). \tag{78}$$

Thus,

$$P^r(x_{(k+1)h}|x_{kh})$$
$$\propto \exp\left(-\frac{\|x_{(k+1)h} - x_{kh} - hb_{kh}^\epsilon(x_{kh})\|^2}{2h\epsilon_{kh}} + \langle \nabla r_{kh}(x_k), x_{(k+1)h} - x_{kh}\rangle + O(h^{3/2})\right) \tag{79}$$
$$\propto \exp\left(-\frac{\|x_{(k+1)h} - x_{kh} - h(b_{kh}^\epsilon(x_{kh}) + \epsilon_{kh}\nabla r_{kh}(x_k))\|^2}{2h\epsilon_{kh}} + O(h^{3/2})\right).$$

$\square$

**Lemma A.5.** *Assume that $(\tilde{x}_t)_{t\in[0,1]}$ satisfies the SDE* (41). *We have that*

$$\int_t^{t'} \tau\chi_\tau \Delta(r \circ X_{\tau,1})(\tilde{x}_\tau)\,\mathrm{d}\tau = \int_t^{t'} \frac{\tau\chi_\tau}{\sqrt{2\epsilon_\tau}}\nabla(r \circ X_{\tau,1})(\tilde{x}_\tau)\cdot\mathrm{d}^-W_\tau - \int_t^{t'} \frac{\tau\chi_\tau}{\sqrt{2\epsilon_\tau}}\nabla(r \circ X_{\tau,1})(\tilde{x}_\tau)\cdot\mathrm{d}W_\tau, \tag{80}$$

*where the forward and backward Itô integrals are defined respectively as*

$$\int_t^{t'} H_\tau \cdot \mathrm{d}W_\tau \;:=\; \lim_{|\pi|\to 0} \sum_{k=0}^{n-1} H_{t_k}\cdot\big(W_{t_{k+1}} - W_{t_k}\big), \tag{81}$$

$$\int_t^{t'} H_\tau \cdot \mathrm{d}^-W_\tau \;:=\; \lim_{|\pi|\to 0} \sum_{k=0}^{n-1} H_{t_{k+1}}\cdot\big(W_{t_{k+1}} - W_{t_k}\big). \tag{82}$$

*Here, $(H_s)_{t\le s\le t'}$ is a process adapted to the filtration induced by the Brownian motion $W$ such that $\mathbb{E}[\int_t^{t'} \|H_s\|^2\,ds] < +\infty$, $\pi = \{t = t_0 < t_1 < \cdots < t_n = t'\}$ is a partition of $[t,t']$ with mesh $|\pi| = \max_k(t_{k+1} - t_k)$, and the limits are $L^2$ limits.*

*Proof.* By definition of the forward and backward Itô integrals,

$$\int_0^t H_\tau\,d^-W_\tau \;-\; \int_0^t H_\tau\,dW_\tau = \lim_{|\pi|\to 0} \sum_{k=0}^{n-1}\big(H_{t_{k+1}} - H_{t_k}\big)\cdot\big(W_{t_{k+1}} - W_{t_k}\big) := [H,W]_t, \tag{83}$$

where $[H,W]_t$ is known as the quadratic variation of $H$ and $W$. Let us set $H_\tau = \frac{\tau\chi_\tau}{\sqrt{2\epsilon_\tau}}\nabla(r \circ X_{\tau,1})(\tilde{x}_\tau) = \gamma_\tau\nabla(r \circ X_{\tau,1})(\tilde{x}_\tau)$, where we defined $\gamma_\tau = \frac{\tau\chi_\tau}{\sqrt{2\epsilon_\tau}}$. By Itô's lemma,

$$d(\gamma_\tau\cdot\nabla(r \circ X_{\tau,1}))(\tilde{x}_\tau)$$

$$= \gamma_t\bigg(\partial_t\nabla(r \circ X_{t,1})(\tilde{x}_\tau) + \nabla^2(r \circ X_{t,1})(\tilde{x}_\tau)\cdot\big(\tilde{b}_t + \epsilon_t s_t - \epsilon_t\nabla r_t\big)(\tilde{x}_\tau) + \epsilon_t\nabla\cdot\nabla^2(r \circ X_{t,1})(\tilde{x}_\tau)\bigg)\,dt$$

$$+ \dot{\gamma}_t\nabla(r \circ X_{t,1})(\tilde{x}_\tau)\,dt + \sqrt{2\epsilon_t}\gamma_t\nabla^2(r \circ X_{t,1})(\tilde{x}_\tau)dW_t. \tag{84}$$

When we simplify the quadratic variation, only the stochastic term survives:

$$[H,W]_t = \lim_{|\pi|\to 0} \sum_{k=0}^{n-1}\sqrt{2\epsilon_{t_k}}\gamma_{t_k}\langle\nabla^2(r \circ X_{t_k,1})(\tilde{x}_{t_k})(W_{t_{k+1}} - W_{t_k}), W_{t_{k+1}} - W_{t_k}\rangle$$

$$= \lim_{|\pi|\to 0} \sum_{k=0}^{n-1}\sqrt{2\epsilon_{t_k}}\gamma_{t_k}\mathrm{Tr}\big(\nabla^2(r \circ X_{t_k,1})(\tilde{x}_{t_k})\big)(t_{k+1} - t_k) = \int_0^t \sqrt{2\epsilon_\tau}\gamma_\tau\Delta(r \circ X_{\tau,1})(\tilde{x}_\tau)\,d\tau$$

$$= \int_0^t \tau\chi_\tau\Delta(r \circ X_{\tau,1})(\tilde{x}_\tau)\,d\tau, \tag{85}$$

which concludes the proof. $\qquad\square$

**Lemma A.6.** *The infinitesimal generator $L_t^\chi$ defined in* (51) *and the HJB residual defined in* (52) *fulfill:*

$$[L_t^\chi e^{r_t}](x) = e^{r_t(x)}\mathcal{R}_t^\chi(x). \tag{86}$$

*Proof.* Applying the Hopf-Cole transformation, we obtain that

$$\partial_t e^{r_t(x)} = e^{r_t(x)}\partial_t r_t(x)$$

$$= e^{r_t(x)}\big(\mathcal{R}_t^\chi(x) - b_t(x)\cdot\nabla r_t(x) - |\chi_t|\big(\|\nabla r_t(x)\|^2 + \Delta r_t(x) + \nabla r_t(x)\cdot s_t(x)\big)\big). \tag{87}$$

And observe that

$$\big(b_t(x) + |\chi_t|s_t(x)\big)\cdot\nabla e^{r_t(x)} + |\chi_t|\Delta e^{r_t(x)}$$

$$= e^{r_t(x)}\big(b_t(x)\cdot\nabla r_t(x) + |\chi_t|\big(\|\nabla r_t(x)\|^2 + \Delta r_t(x) + \nabla r_t(x)\cdot s_t(x)\big)\big). \tag{88}$$

Plugging this into (87) yields

$$\partial_t e^{r_t(x)} = e^{r_t(x)}\mathcal{R}_t^\chi(x) - (b_t(x) + |\chi_t|s_t(x)) \cdot \nabla e^{r_t(x)} - |\chi_t|\Delta e^{r_t(x)}. \tag{89}$$

And plugging the expression of the infinitesimal generator in (51) into (89) yields

$$0 = e^{r_t(x)}\mathcal{R}_t^\chi(x) - [L_t^\chi e^{r_t}](x). \tag{90}$$

$\square$

# B ANALYZING THE PERFORMANCE OF TEST-TIME SAMPLING ALGORITHMS

## B.1 SIMULATING THE DYNAMICS WITH SMC

The natural approach to handle the weights $e^{A_t}$ in Proposition 2.2 and in Proposition A.1 is sequential Monte Carlo (SMC), which is implemented in Algorithm 1. Let $\mathcal{T} = (t_k)_{k=0}^K$ be an annealing schedule satisfying $0 = t_0 < \cdots < t_K = 1$. The SMC sampling procedure starts with $N$ particles $\mathbf{X}_0 = (X_0^n)_{n\in[N]}$ drawn from the reference, $X_0^n \sim \rho_0$, and initial weights $\mathbf{w}_0 = (w_0^n)_{n\in[N]}$ with $w_0^n = 1$. For each subsequent iteration $k \in [K]$, we produce $\mathbf{X}_k$ and $\mathbf{w}_k$ by propagating, reweighting, and optionally resampling[†]. In what follows, we use a notation similar to the one of Syed et al. (2024).

**Propagate.** Evolve $\mathbf{X}_{k-1}$ forward with the Markov transition kernel $M_{t_{k-1},t_k}(x_{k-1})$ to obtain $\mathbf{X}_k = (X_k^n)_{n\in[N]}$, where

$$X_k^n \sim M_{t_{k-1},t_k}(X_{k-1}^n). \tag{91}$$

For the dynamics of Proposition 2.2 and Proposition A.1, the Markov transition kernels are the Euler-Maruyama updates for the SDEs (13) and (35), respectively:

$$X_k^n = X_{k-1}^n + [b_{t_{k-1}}(X_{k-1}^n) + \epsilon_{t_{k-1}}(s_{t_{k-1}}(X_{k-1}^n) + \nabla r_{t_{k-1}}(X_{k-1}^n))](t_k - t_{k-1})$$
$$+ \sqrt{2\epsilon_k(t_k - t_{k-1})}\xi_{k-1}^n, \quad \xi_{k-1}^n \sim N(0, \mathrm{I}), \tag{92}$$

$$X_k^n = X_{k-1}^n + [b_{t_{k-1}}(X_{k-1}^n) + \chi_{t_{k-1}}\nabla r_{t_{k-1}}(X_{k-1}^n) + \epsilon_{t_{k-1}}(s_{t_{k-1}}(X_{k-1}^n) + \nabla r_{t_{k-1}}(X_{k-1}^n))](t_k - t_{k-1})$$
$$+ \sqrt{2\epsilon_k(t_k - t_{k-1})}\xi_{k-1}^n, \quad \xi_{k-1}^n \sim N(0, \mathrm{I}), \tag{93}$$

**Reweight.** Update the weights using the incremental weight function $g_{t_{k-1},t_k}(x_{k-1}, x_k)$:

$$\mathbf{w}_k = (w_k^n)_{n\in[N]}, \qquad w_k^n = w_{k-1}^n\, g_{t_{k-1},t_k}(X_{k-1}^n, X_k^n). \tag{94}$$

The incremental weight functions $g_{t_{k-1},t_k}(X_{k-1}^n, X_k^n)$ and weights $w_k^n$ can be computed using the framework of Appendix A.4:

$$g_{t_{k-1},t_k}(X_{k-1}^n, X_k^n) = \exp\left(A_k^n - A_{k-1}^n\right), \qquad w_k^n = \exp\left(A_k^n\right). \tag{95}$$

Here, the difference $A_k^n - A_{k-1}^n$ is computed for each particle using one of the approaches described in Appendix A.4.

**Resample (optional).** On a (possibly random) subset of iterations $\mathcal{R} \subseteq [K]$ determined by a criterion depending on $(\mathbf{X}_k, \mathbf{w}_k)$, apply a resampling step

$$\mathbf{X}_k \leftarrow \mathrm{resample}(\mathbf{X}_k, \mathbf{w}_k),$$

to stabilize $\mathbf{w}_k$ and favor propagation of particles with higher relative weight. Concretely, set $X_k^n \leftarrow X_k^{a_k^n}$, where $a_k = (a_k^n)_{n\in[N]}$ is a random ancestor index vector with $a_k^n \in [N]$ and

$$\mathbb{P}(a_k^n = m \mid \mathbf{w}_k) = \frac{w_k^m}{\sum_{j\in[N]} w_k^j}, \qquad m \in [N].$$

---

[†]While in Algorithm 1 we allow for a number of clones $C$ greater than one, for simplicity, the analysis we perform in this section is with $C = 1$.

After resampling, reset the weights via $w_k^n \leftarrow 1$ for all $n$. See (Chopin et al., 2022) for annealed-SMC-specific resampling schemes, and (Dai et al., 2020) for a recent review of SMC samplers.

The SMC procedure yields an unbiased estimate of the expectation $\int_{\mathbb{R}^d} h(x)\hat{\rho}_{t_k}(x)dx$ for any test function $h$ and any $k \in [K]$. Namely, if we let $\bar{\rho}_t$ be the unnormalized density as defined in Appendix A.1, recall that $\mathcal{R} \cap [k]$ denotes the subset of iterations where a resampling step happens, and for $r \in \mathcal{R}$ we let $w_r$ be the weight prior to resetting to 1, we have that

$$\int_{\mathbb{R}^d} h(x)\bar{\rho}_{t_k}(x)dx = \mathbb{E}\left[\left(\prod_{r \in \mathcal{R} \cap [k-1]} \frac{1}{N}\sum_{n=1}^N w_r^n\right)\frac{1}{N}\sum_{n=1}^N w_k^n h(X_k^n)\right], \tag{96}$$

Setting $h \equiv 1$ and recalling that $\hat{\rho}_{t_k}(x) = \bar{\rho}_{t_k}(x)/\int_{\mathbb{R}^d} \bar{\rho}_{t_k}(x)dx$, we obtain that

$$\int_{\mathbb{R}^d} h(x)\hat{\rho}_{t_k}(x)dx = \frac{\mathbb{E}\left[\hat{Z}_{\mathrm{SMC}}^{(k)} \frac{\frac{1}{N}\sum_{n=1}^N w_{t_k}^n h(X_{t_k}^n)}{\frac{1}{N}\sum_{n=1}^N w_{t_k}^n}\right]}{\mathbb{E}\left[\hat{Z}_{\mathrm{SMC}}^{(k)}\right]}, \quad \hat{Z}_{\mathrm{SMC}}^{(k)} = \left(\prod_{r \in \mathcal{R} \cap [k-1]} \frac{1}{N}\sum_{n=1}^N w_r^n\right) \times \left(\frac{1}{N}\sum_{n=1}^N w_k^n\right). \tag{97}$$

If we set $k = K$, we obtain that

$$\int_{\mathbb{R}^d} h(x)\hat{\rho}_1(x)dx = \frac{\mathbb{E}\left[\hat{Z}_{\mathrm{SMC}} \frac{\frac{1}{N}\sum_{n=1}^N w_K^n h(X_K^n)}{\frac{1}{N}\sum_{n=1}^N w_K^n}\right]}{\mathbb{E}\left[\hat{Z}_{\mathrm{SMC}}\right]}, \qquad \text{where } \hat{Z}_{\mathrm{SMC}} = \prod_{r \in \mathcal{R}} \frac{1}{N}\sum_{n=1}^N w_r^n. \tag{98}$$

$\hat{Z}_{\mathrm{SMC}}$ is known as the SMC normalization constant. Observe that $\mathbb{E}[\hat{Z}_{\mathrm{SMC}}] = \int_{\mathbb{R}^d} \bar{\rho}_1(x)dx := Z$. Thus, low $\mathrm{Var}[\hat{Z}_{\mathrm{SMC}}/Z]$ is a proxy for good performance of the SMC procedure. In Appendix B.3 we reproduce a result by Syed et al. (2024) that expresses $\mathrm{Var}[\hat{Z}_{\mathrm{SMC}}/Z]$ in terms of the parameters of the SMC procedure, using the concept of total discrepancy from Appendix B.2.

## B.2 THE INCREMENTAL AND TOTAL DISCREPANCIES

Following Syed et al. (2024), we define the normalized incremental weight function $G_{t,t'}$ as

$$G_{t,t'}(x, x') = \frac{g_{t,t'}(x, x')}{\mathbb{E}_{(X,X')\sim\hat{\rho}_t\otimes M_{t,t'}}[g_{t,t'}(X, X')]} \tag{99}$$

where $g_{t,t'}$ is the unnormalized incremental weight function defined in (94) and $M_{t,t'}$ is the Markov transition kernel defined in (91).

Given $G_{t,t'}$ from time $t$ to time $t'$, the *incremental discrepancy* $D(t, t')$ is defined as

$$D(t, t') = \log\left(1 + \mathrm{Var}_{(X,X')\sim\hat{\rho}_t\otimes M_{t,t'}}\left(G_{t,t'}(X, X')\right)\right). \tag{100}$$

Given a sequence of timesteps $\mathcal{T} = (t_k)_{k=0}^K$ with $t_0 = 0$, $t_K = 1$, tor $k \leq k'$, define $D(\mathcal{T}, t_k, t_{k'})$ as the *accumulated discrepancy* between iterations $k$ and $k'$ and $D(\mathcal{T})$ as the *total discrepancy*:

$$D(\mathcal{T}, t_k, t_{k'}) = \sum_{k''=k+1}^{k'} D(t_{k''-1}, t_{k''}), \qquad D(\mathcal{T}) = D(\mathcal{T}, 0, 1). \tag{101}$$

Observe that the incremental discrepancy can be expressed in terms of the first and second moments of $g_{t,t'}(X, X')$:

$$D(t, t') = \log\left(1 + \mathrm{Var}_{(X,X')\sim\hat{\rho}_t\otimes M_{t,t'}}\left[\frac{g_{t,t'}(X, X')}{\mathbb{E}_{(X'',X''')\sim\hat{\rho}_t\otimes M_{t,t'}}[g_{t,t'}(X'', X''')]}\right]\right)$$

$$= \log\left(1 + \frac{\mathbb{E}_{(X,X')\sim\hat{\rho}_t\otimes M_{t,t'}}[g_{t,t'}(X, X')^2]}{\mathbb{E}_{(X'',X''')\sim\hat{\rho}_t\otimes M_{t,t'}}[g_{t,t'}(X'', X''')]^2} - \frac{\mathbb{E}_{(X,X')\sim\hat{\rho}_t\otimes M_{t,t'}}[g_{t,t'}(X, X')]^2}{\mathbb{E}_{(X'',X''')\sim\hat{\rho}_t\otimes M_{t,t'}}[g_{t,t'}(X'', X''')]^2}\right)$$

$$= \log\mathbb{E}_{(X,X')\sim\hat{\rho}_t\otimes M_{t,t'}}[g_{t,t'}(X, X')^2] - 2\log\mathbb{E}_{(X,X')\sim\hat{\rho}_t\otimes M_{t,t'}}[g_{t,t'}(X, X')] \tag{102}$$

We want to obtain a consistent estimator for $D(t, t')$. Following (Syed et al., 2024, Sec. 5.1), if we are using a single SMC run with $N$ particles, and we let $g_k^n = g_{t_{k-1}, t_k}(X_{k-1}^n, X_k^n)$, then the following estimator is consistent:

$$\hat{D}_k = \log \hat{g}_{k,2} - 2 \log \hat{g}_{k,1} - \log \hat{g}_{k,0}, \quad \text{where for } i \in \{0, 1, 2\}, \quad \hat{g}_{k,i} = \sum_{n \in [N]} w_{k-1}^n (g_k^n)^i. \tag{103}$$

To get a consistent estimator using $N_R$ SMC runs, each with $N$ particles, we compute the normalization constant $\hat{Z}_{\text{SMC}}^{(k,j)}$ in (97) for each SMC run $j = 1 : N_R$, and define $\hat{D}_k$ as in (103), but where $\hat{g}_{k,i}$ takes the form

$$\hat{g}_{k,i} = \sum_{j=1}^{N_R} \hat{Z}_{\text{SMC}}^{(k,j)} \frac{\sum_{n \in [N]} w_{k-1}^{(n,j)} (g_k^{(n,j)})^i}{\sum_{n \in [N]} w_{k-1}^{(n,j)}}, \qquad \text{where} \quad g_k^{(n,j)} = g_{t_{k-1}, t_k}(X_{k-1}^{(n,j)}, X_k^{(n,j)}), \tag{104}$$

and $X_k^{(n,j)}, w_k^{(n,j)}$ is the $n$-th particle and weight of $j$-th SMC run at iteration $k$.

### B.3 THE VARIANCE OF THE SMC NORMALIZATION CONSTANT

The total discrepancy defined in equation (101) is related to the variance of the SMC normalization constant $Z_{\text{SMC}}$ via the following result, which was proven by Syed et al. (2024) as a generalization of a result of Dai et al. (2020):

**Theorem B.1** (Theorem 1, Syed et al. (2024)). *Suppose that the following assumptions on the normalized incremental weights $G_k^n = G_{t_{k-1}, t_k}(X_{k-1}^n, X_k^n)$ defined in (99) hold:*

- Assumption 1 (Integrability). *For all $n \in [N]$, $k \in [K]$, $G_k^n$ has finite variance with respect to $\hat{\rho}_{t_k} \otimes M_{t_{k-1}, t_k}$.*

- Assumption 2 (Temporal indep.). *For $n \in [N]$, $(G_k^n)_{t \in [T]}$ are independent.*

- Assumption 3 (Particle indep.). *For $k \in [K]$, $(G_k^n)_{n \in [N]}$ are independent.*

- Assumption 4 (Efficient local moves). *For each $n \in [N]$ and $k \in [k]$,*

$$G_k^n \overset{d}{=} G_{t_{k-1}, t_k}(X_{k-1}, X_k), \qquad (X_{k-1}, X_k) \sim \pi_{t_{k-1}} \otimes M_{t_{k-1}, t_k}.$$

*Assume also that $N > 1$, $D(\mathcal{T}) > 0$. For every resample schedule $\mathcal{T}_R = (t_r)_{r=0}^R$, there exists a unique $1 \le R_{\text{eff}} \le \mathbb{E}[R]$ such that*

$$\text{Var}\left(\frac{\hat{Z}_{\text{SMC}}}{Z}\right) = \frac{1}{N}\left(\exp\left(\frac{D(\mathcal{T})}{R_{\text{eff}}}\right) - 1\right) R_{\text{eff}} - 1. \tag{105}$$

*Moreover, $R_{\text{eff}} = 1$ if and only if $D(\mathcal{T}, t_{r-1}, t_r) \overset{\text{a.s.}}{=} D(\mathcal{T})$ for some $r \in [R]$, and $R_{\text{eff}} = \mathbb{E}[R]$ if and only if $R$ is a.s. constant and $D(\mathcal{T}, t_{r-1}, t_r) \overset{\text{a.s.}}{=} D(\mathcal{T})/R$ for some $r \in [R]$.*

**Remark B.2.** *As remarked by Syed et al. (2024), Assumptions 1–4 constitute an idealized model similar to the one considered by other works in the area (Grosse et al., 2013; Dai et al., 2020). While Assumption 1 is weak, Assumptions 2–4 are not. Assumption 3 only holds when no resampling is performed, and Assumptions 2–4 only hold (approximately) when a number of MCMC steps are interleaved with the SMC updates. However, Syed et al. (2024, Sec. 6.1) show that empirically, the scaling (105) is consistent with empirical observation.*

### B.4 OPTIMIZING THE ANNEALING SCHEDULE TO MINIMIZE THE TOTAL DISCREPANCY

As defined in (101), the total discrepancy depends not only on the continuous time dynamics for positions and weights, but also on the specific annealing schedule $\mathcal{T} = (t_k)_{k=0}^K$. For a fixed $K$, it is possible to characterize and find the annealing schedule that minimizes the total discrepancy.

Under technical regularity assumptions (see (Syed et al., 2024, Sec. 4.1)), the incremental discrepancy admits the asymptotic expansion

$$G_{t,t+\Delta t} = 1 + S_t\,\Delta t + o(\Delta t),$$

and hence the local changes and variance of the incremental discrepancy $G_{t,t+\Delta t}$ are encoded in $S_t$ and its variance $\delta(t)$, defined as

$$S_t = \left.\frac{\partial}{\partial t'}G_{t,t'}\right|_{t'=t}, \qquad \delta(t) = \mathrm{Var}_{t,t}[S_t].$$

Using this expansion, we can expand the incremental discrepancy as follows:

$$D(t, t + \Delta t) = \delta(t)\Delta t + O(\Delta t^3). \tag{106}$$

**Scheduler generators**  A *schedule generator* is a continuously twice-differentiable function $\varphi :$ $[0, 1] \to [0, 1]$ such that $\varphi(0) = 0$, $\varphi(1) = 1$, and $\dot\varphi(u) = \frac{d}{du}\varphi(u) > 0$. Given $K \in \mathbb{N}$, $\varphi$ generates an annealing schedule $\mathcal{T} = (t_k)_{k=0}^K$ where

$$t_k = \varphi(u_k), \qquad u_k = \frac{k}{K}.$$

In the following, without loss of generality, we restrict our attention to schedules generated by schedule generator.

By the mean value theorem, we have

$$t_k - t_{k-1} \approx \frac{\dot\varphi(u_k)}{K}.$$

Combining this with equation (106), we obtain

$$D(t_{k-1}, t_k) \approx \frac{\delta(\varphi(u_k))\,\dot\varphi(u_k)^2}{K^2}. \tag{107}$$

By summing over $k$ and using Riemann approximations, we can approximate $D(\mathcal{T}, t_k, t_{k'})$ and $D(\mathcal{T})$ in terms of $E(\varphi, u_{t_k}, u_{t_{k'}})$ and $E(\varphi)$, defined as the integral of $\delta(\varphi(u))\,\dot\varphi(u)^2$,

$$E(\varphi, u, u') = \int_u^{u'} \delta(\varphi(v))\,\dot\varphi(v)^2\,dv, \qquad E(\varphi) = E(\varphi, 0, 1).$$

**Proposition B.3** (Proposition 1, Syed et al. (2024)). *Suppose Assumptions 5 to 8 hold. There exists* $C_D(\varphi) > 0$ *such that, for $k, k' \in [K]$,*

$$\left|D(\mathcal{T}, t_k, t_{k'}) - \frac{1}{K}E(\varphi, u_k, u_{k'})\right| \le \frac{C_D(\varphi)\,|t_{k'} - t_k|}{T^3}.$$

An immediate consequence of Proposition 1 is that, in the dense schedule limit as $K \to \infty$, the total discrepancy $D(\mathcal{T})$ is asymptotically equivalent to $\frac{E(\varphi)}{K}$. Hence, for a fixed $K$, optimizing $D(\mathcal{T})$ with respect to $\mathcal{T}$ is asymptotically equivalent to the following problem:

$$\min_{\varphi:[0,1]\to[0,1]} \int_0^1 \delta(\varphi(u))\,\dot\varphi(u)^2\,du, \quad \text{s.t.} \quad \int_0^1 \dot\varphi(u)\,du = 1. \tag{108}$$

Jensen's inequality implies that

$$\int_0^1 \delta(\varphi(u))\,\dot\varphi(u)^2\,du \ge \left(\int_0^1 \sqrt{\delta(\varphi(u))}\,\dot\varphi(u)\,du\right)^2, \tag{109}$$

with equality if and only if there exists a constant $\Lambda > 0$ such that $\sqrt{\delta(\varphi^\star(u))}\,\dot\varphi^\star(u) = \Lambda$ for $u$ a.e. in $[0, 1]$. Defining

$$\Lambda(t) = \int_0^t \sqrt{\delta(u)}\,du, \tag{110}$$

by the chain rule, we have equivalently that

$$\Lambda = \Lambda'(\varphi^\star(t))\,\dot{\varphi}^\star(t) = \frac{d}{dt}\Lambda(\varphi^\star(t)) \implies \Lambda(\varphi^\star(t)) = \Lambda t \implies \varphi^\star(t) = \Lambda^{-1}(\Lambda t). \qquad (111)$$

Setting $t = 1$ in $\Lambda(\varphi^\star(t)) = \Lambda t$ also implies that $\Lambda = \Lambda(\varphi^\star(1)) = \Lambda(1) = \int_0^1 \sqrt{\delta(u)}\,du$. Syed et al. (2024) refer to $\Lambda(t)$ and $\Lambda$ as the *local barrier* and the *global barrier* associated to the SMC algorithm. We refer to $\Lambda$ as the **thermodynamic length** associated to the algorithm.

A change of the integration variable implies that for any schedule generator $\varphi$,

$$\Lambda(\varphi(t)) = \int_0^{\varphi(t)} \sqrt{\delta(u)}\,du = \int_0^t \sqrt{\delta(\varphi(u))}\dot{\varphi}(u)\,du,$$

$$\implies \Lambda(t_k) = \Lambda(\varphi(u_k)) = \int_0^{u_k} \sqrt{\delta(\varphi(u))}\dot{\varphi}(u)\,du \approx \sum_{k'=1}^{k} \frac{\sqrt{\delta(\varphi(u_k))}\,\dot{\varphi}(u_k)}{K} = \sum_{k'=1}^{k} \sqrt{D(t_{k-1}, t_k)},$$

$$\Lambda = \int_0^1 \sqrt{\delta(\varphi(u))}\dot{\varphi}(u)\,du \approx \sum_{k=1}^{K} \sqrt{D(t_{k-1}, t_k)}.$$

$$(112)$$

where the last equality holds by (107). This allows us to approximate the local and global barriers using (102) to compute the incremental discrepancy $D(t_{k-1}, t_k)$. Once we have an estimate $\hat{\Lambda}$ of the barrier, Syed et al. (2024) propose to iteratively refine the annealing schedule by resetting $t_k \leftarrow \hat{\Lambda}^{-1}(\hat{\Lambda}k/K)$.

Observe that given the quantities $\sum_{k'=1}^{k} D(t_{k'-1}, t_{k'})$ and $\sum_{k'=1}^{k} \sqrt{D(t_{k'-1}, t_{k'})}$, we have that

$$\sum_{k'=1}^{k} D(t_{k'-1}, t_{k'}) = D(\mathcal{T}, 0, t_{k'}) \approx \frac{1}{K}E(\varphi, u_0, u_{k'}) \geq \frac{1}{K}\Lambda(t_k)^2 \approx \frac{1}{K}\left(\sum_{k'=1}^{k} \sqrt{D(t_{k'-1}, t_{k'})}\right)^2, \qquad (113)$$

Thus, the quantity

$$\frac{\left(\sum_{k'=1}^{k} \sqrt{D(t_{k'-1}, t_{k'})}\right)^2}{K\sum_{k'=1}^{k} D(t_{k'-1}, t_{k'})} \qquad (114)$$

should fall within $[0, 1]$ and close to 1 when the annealing schedule $\mathcal{T}$ is close to the optimal one.

## C  IMPLEMENTATION DETAILS

The complete pseudocode of FMTT is given in Algorithm 1.

---

**Algorithm 1** Inference-time adaptation of flow maps

---

1: **Input**: # simulation steps $K$, # resampling steps $R$, # particles $N$, # particle clones $C$, time sequence $(t_k)_{k=0:K}$, sequences $(\epsilon_k)_{k=0:K}$ and $(\chi_k)_{k=0:K}$

2: **for** $i = 1 : N$, initialize $x_i^0 = N(0, \mathrm{I})$ i.i.d. Let $X^0 = (x_i^0)_{i=1:N}$.

3: Clone the particles: $\bar{X}^0 = (x_{ij}^0)_{i=1:N, j=1:C}$, where $(x_{ij}^0)^{j=1:C}$ are equal copies of $x_i^0$.

4: **if** Sampling **then for** $i = 1 : N$ and $j = 1 : C$, initialize $A_{ij}^0 = 0$.

5: **for** $k = 0 : K - 1$ **do**

6: $\quad \Delta t \leftarrow t_{k+1} - t_k$

7: $\quad$ **for** $i = 1 : N$ and $j = 1 : C$ **do** $\qquad\qquad\qquad\qquad\qquad$ ▷ Update particles

8: $\quad\quad x_{ij}^{k+1} = x_{ij}^k + [v_{t_k, t_k}(x_{ij}^k) + \chi_k \nabla r_{t_k}(x_{ij}^k) + \epsilon_k(s_{t_k}(x_{ij}^k) + \nabla r_{t_k}(x_{ij}^k))]\Delta t + \sqrt{2\epsilon_k \Delta t}\xi_{ij}^k, \quad \xi_{ij}^k \sim N(0, \mathrm{I})$.

9: $\quad\quad$ **if** Sampling **then**

10: $\quad\quad\quad$ Compute $A_{ij}^{k+1}$ from $A_{ij}^k$ using approach *(i)* (66), approach *(ii)* (68) or approach *(iii)* (70)-(72) from Appendix A.4. Note that simplifications occur when $\xi_k \equiv 0$ (Remark A.3).

11: $\quad\quad$ **end if**

12: $\quad$ **end for**

13: $\quad$ **if** $k = 0 \pmod{K/R}$ and $k > 0$ **then** $\qquad\qquad$ ▷ Resample / select particles

14: $\quad\quad$ **if** Sampling **then**

15: $\quad\quad\quad$ Define probabilities $p^k = \text{softmax}(A^k) = \left( \exp(A_{ij}^k) / \sum_{i'j'} \exp(A_{i'j'}^k) \right)_{i'=1:N, j'=1:C}$

16: $\quad\quad\quad$ Resample $X^k = (x_i^k)_{i=1:N} \sim \sum_{i'=1}^n \sum_{j'=1}^C p_{i'j'}^k \delta_{x_{i'j'}^k}$, i.i.d., or using Quasi-Monte Carlo

17: $\quad\quad\quad$ Set $A_{ij}^k = 0$.

18: $\quad\quad$ **else if** Searching **then**

19: $\quad\quad\quad$ Select $X^k = (x_i^k)_{i=1:N}$ as the top-$n$ samples among $\bar{X}^k$ with respect to $r_{t_k}(x_{ij}^k)$.

20: $\quad\quad$ **end if**

21: $\quad\quad$ Clone the particles: $\bar{X}^k \leftarrow (x_{ij}^k)_{i=1:N}^{j=1:C}$, where $(x_{ij}^k)^{j=1:C}$ are equal copies of $x_i^k$.

22: $\quad$ **end if**

23: **end for**

24: **return** $x$

---

## D   MNIST EXPERIMENTS

We ran the MNIST experiments using a flow map model that we trained from scratch, with stochastic interpolants $\alpha_t = 1 - t$, $\beta_t = t$. We use the unconditional flow map, which samples the distribution $\rho_1$ of all MNIST digits. We run Algorithm 1 with $N = 128$ particles and $K = 200$ simulation steps.

We ablate the following look-ahead approaches introduced in Section 2.2:

- Naive look-ahead / no look-ahead (NL): $r_t(x) = tr(x)$.
- Denoiser look-ahead (WD): $r_t(x) = tr(\mathsf{D}_t(x))$.
- Flow map look-ahead (FMTT): $r_t(x) = tr(X_{t,1}(x))$.

We set $\epsilon_t = \alpha_t = 1 - t$. We consider the four choices of $\chi_t$ described in Appendix A.3:

- Default: $\chi_t = 0$.
- Tilted score dynamics: $\chi_t = \eta_t := \alpha_t(\frac{\dot{\beta}_t}{\beta_t}\alpha_t - \dot{\alpha}_t) = (1 - t)\big(\frac{1-t}{t} + 1\big) = \frac{1-t}{t}$. In practice, we add an offset of $0.05$ to $\beta_t$ in the denominator: $\chi_t = \alpha_t(\frac{\dot{\beta}_t}{\beta_t+0.05}\alpha_t - \dot{\alpha}_t) = (1 - t)\big(\frac{1-t}{t+0.05} + 1\big) = \frac{1.05\times(1-t)}{t+0.05}$.
- Local tilt dynamics: $\chi_t = \epsilon_t = 1 - t$.
- Base dynamics: $\chi_t = -\epsilon_t = -1 + t$.

We run Sampling and Searching experiments.

- For Sampling experiments (Figure 7), we set $r(x) = 0.1 \times \log p_\theta(0|x)$, where $p_\theta$ is a pre-trained MNIST classifier and $\log p_\theta(0|x)$ is the probability that the classifier assigns to the digit $0$ for the image $x$. Following Section 2, the target distribution that we aim to sample is the tilted probability distribution is $\hat{\rho}_1(x) \propto \rho_1(x)\exp\big(0.1 \times \log p_\theta(0|x)\big)$. We run the sampling experiments with $C = 1$ clone. We perform the log-weight update in Line 10 using approach *(iii)* (see Appendix A.4.1 for a discussion on the different approaches). We remark again that the choice $\chi_t = 0$ is much faster computationally, as it does not require estimating expectations. For the other three choices of $\chi_t$, we use $M = 400$ noisy samples when computing (70) or (72). Instead of resampling every $R$ steps with $R$ fixed, we resample when the effective sample size, which is computed as $\sum_{i=1}^{N}\exp(A_i^k)^2/\big(\sum_{i=1}^{N}\exp(A_i^k)\big)^2$, falls below $0.85 \times N$.
- For Searching experiments (Figure 8), $r(x) = 0.05 \times \log p_\theta(0|x)$. Note that in this case, there does not exist an explicit form for the target probability distribution, which is different for each algorithm. We run the simulation with $C = 2$ clones, and do one resampling step after the 100th simulation step (in the middle of the generation).

**Sampling results**   In Figure 7, we show samples generated using each algorithm and plot four metrics. For each algorithm, the metrics were computed from 16 runs, each with $N = 128$ particles, and the bars show the average value plus/minus the standard error.

- Average reward: this is the expected value of $\log p_\theta(0|x)$ for the samples generated with the algorithm (without the multiplier $0.1$). We also print the ground truth value, which we compute by sampling $400 \times 128 = 51200$ independent samples from the unconditional model, and reweighting them by the factor $\exp\big(0.1 \times \log p_\theta(0|x)\big)$ upon normalization.
- Class entropy: this is the average value of the entropy of the probability distribution $(p_\theta(i|x))_{i=0}^{9}$ induced by the classifier $p_\theta$ for each generated sample $x$, i.e. $-\sum_{i=0}^{9}\log p_\theta(i|x)$. We also print the ground truth value, which we compute by sampling $51200$ samples and reweighting them accordingly.
- Total discrepancy: this is the quantity $D(\mathcal{T}) = \sum_{k=1}^{K}D(t_{k-1}, t_k)$ defined in equation (101), where $D(t_{k-1}, t_k)$ are computed using the procedure described in Appendix B.2.
- Thermodynamic length: this is the $\Lambda = \sum_{k=1}^{K}\sqrt{D(t_{k-1}, t_k)}$ defined in (112) (strictly speaking, this is an approximation of the actual thermodynamic length).

We observe that the only $\chi_t$ choice for which the ground truth values for the average reward and the class entropy fall within the error bars is $\chi_t = 0$. For this choice, the look-ahead approach that results in lower total discrepancy and thermodynamic length is FMTT. To understand why this is the case, recall that the evolution of the log-weights for $\chi_t = 0$ is given by the ODE $\frac{dA_t}{dt} = \frac{d}{ds}r_s(X_s^0)\big|_{X_t^0=\tilde{x}_t, s=t}$. Observe that when $r_t(x) = r_t^\star(x) := r(X_{t,1}(x))$, then $\frac{dA_t}{dt} = 0$, which implies that the total discrepancy and the thermodynamic length are zero, and it is easy to see that this is the only choice of $r_t$ for which they are zero[‡]. Thus, the total discrepancy and the thermodynamic length values can be regarded as a measure of how close a given $r_t$ is to $r_t^\star :=$ $r \circ X_{t,1}$. From their values in the first row of Figure 7, we deduce that the flow map look-ahead reward $r_t(x) = tr(X_{t,1}(x))$ is the closest one to $r_t^\star(x) = r(X_{t,1}(x))$, which is unsurprising because they only differ by a factor $t$.

For all other $\chi_t$ choices, the ground truth class entropy values fall outside of the error bars of the model. Moreover, for these other $\chi_t$ choices, the total discrepancy and thermodynamic length are substantially higher for FMTT than for NL and WD, and are lowest for WD. To understand why this is the case, observe that the evolution of the log-weights for $\chi_t = \eta_t$ and $\chi_t = \epsilon_t$ is given by the ODE $\frac{dA_t}{dt} = \frac{d}{ds}\mathbb{E}[e^{r_s(X_s^\chi)-r_t(\tilde{x}_t)} \,|\, X_t^\chi = \tilde{x}_t]\big|_{s=t}$. Note that when $r_t$ is the optimal time-dependent reward in the stochastic optimal control sense, i.e., $r_t(x) = r_t^\star(x) := \log \mathbb{E}[\exp(r_1(X_1^\chi))|X_t^\chi = \tilde{x}_t]$[§], we have that $\mathbb{E}[e^{r_s(X_s^\chi)-r_t(\tilde{x}_t)} \,|\, X_t^\chi = \tilde{x}_t] = 0$ for all $0 \le t \le s \le 1$. Hence, for this choice of $r_t$, the total discrepancy and the thermodynamic length are zero, and it is the only choice for which they are zero. Like before, the total discrepancy and the thermodynamic length values can be regarded as a measure of how close a given $r_t$ is to $r_t^\star$. From their values in the second and third row of Figure 7, we deduce that the denoiser look-ahead reward $r_t(x) = tr(\mathsf{D}_t(x))$ is the closest one to $r_t^\star(x)$, and $r_t(x) = tr(X_{t,1}(x))$ is the farthest apart.

**Searching results**   In Figure 8, we show samples generated using each algorithm and plot the average reward and average class entropy methods, which are also computed from 16 runs with $N = 128$ particles each, as in the sampling case. In this case, there are no ground truth values to compare to. We observe that all else equal, FMTT with $\chi_t = \eta_t$ is the most effective choice in that it achieves the highest average reward and the lowest class entropy, hence sampling the digit 0 more often. Accordingly, FMTT with $\chi_t = \eta_t$ is the choice we make for our high dimensional search experiments throughout the paper.

## E   ADDITIONAL VLM-AS-JUDGE EXAMPLES

As described in the paper, we use Qwen2.5-VL-7B-Instruct to define rewards expressed as yes/no questions over one or more context images. This makes it possible to cast diverse objectives as test-time search problems, including style consistency, character consistency, and multi-subject generation.

Here, we demonstrate the style consistency case. The VLM receives both a reference image and a generated image and is asked whether they share the same art style. FMTT then optimizes this reward, producing generations more closely aligned with the reference style. Qualitative results are shown in Figure 9.

[‡]Observe that $r_t(x) = r_t^\star(x) = r(X_{t,1}(x))$ cannot be used in practice because we require that $r_0(x)$ is constant so that we can sample from $\hat{\rho}_0$

[§]This holds by the path integral characterization of stochastic optimal control.

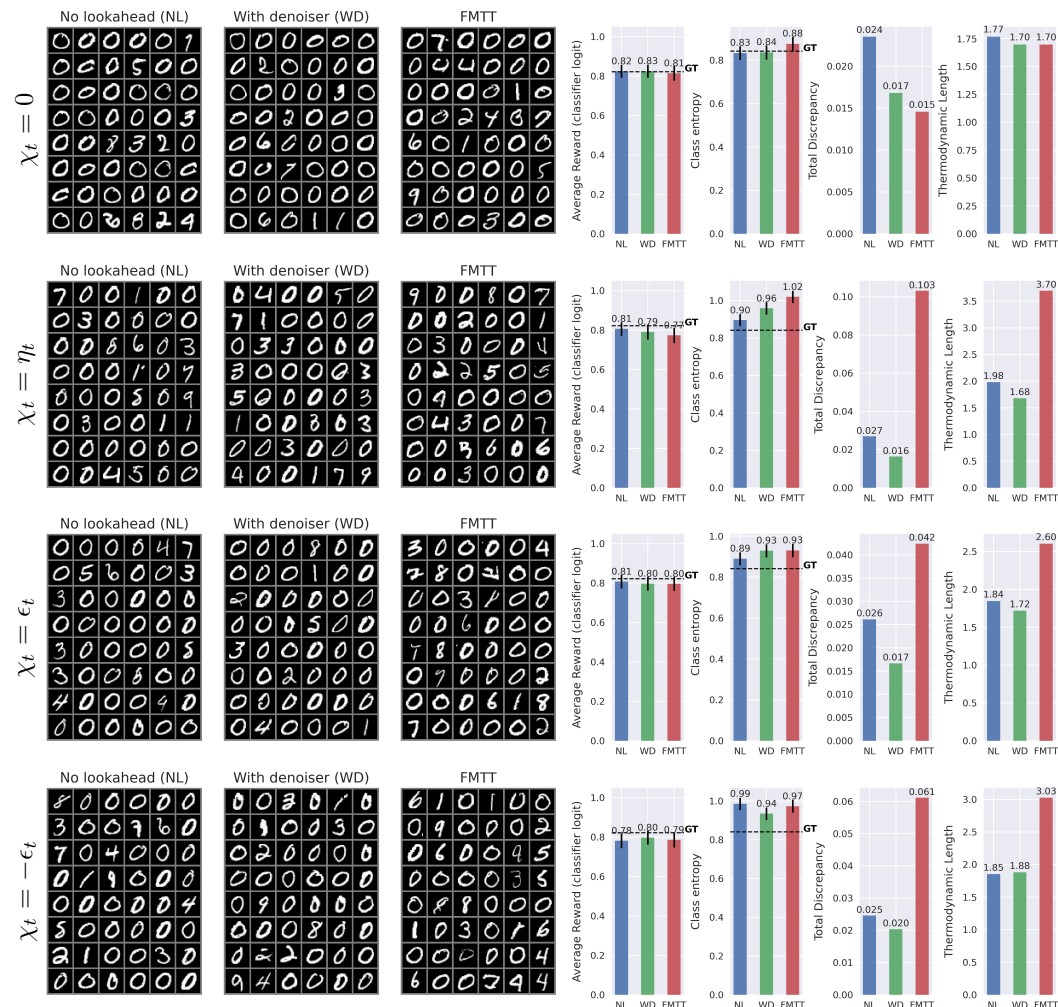

**Figure 7:** Comparison of MNIST tilted sampling to generate digits that would be classified as zeros: the reward is $r(x) = 0.1 \times \log p(0|x)$. The four rows correspond to the dynamics with different $\chi_t$ choices described in Appendix A.3.

# F  VLM REWARD HACKING

As discussed in the paper, a challenge of using VLMs (or any non-verifiable reward model) is the risk of the search process exploiting loopholes. This happens when the algorithm produces images that either act as adversarial examples for the VLM or satisfy the literal question without achieving the intended effect. Figure 10 shows such a case.

**Figure 8:** Comparison of MNIST greedy search to generate digits that would be classified as zeros: the reward is $r(x) = 0.05 \times \log p(0|x)$. The four rows correspond to the dynamics with different $\chi_t$ choices described in Appendix A.3.

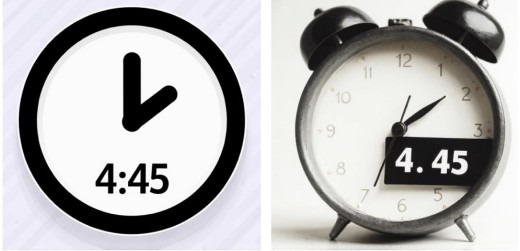

*Prompt: "An analog clock showing exactly 4:45"*
*VLM question: "Is the analog clock showing 4:45?"*

**Figure 10:** VLM reward hacking. Instead of the clock being at 4:45, the search process finds a way to "cheat" by writing the text 4:45 on the clock face and achieving high rewards from the VLM.

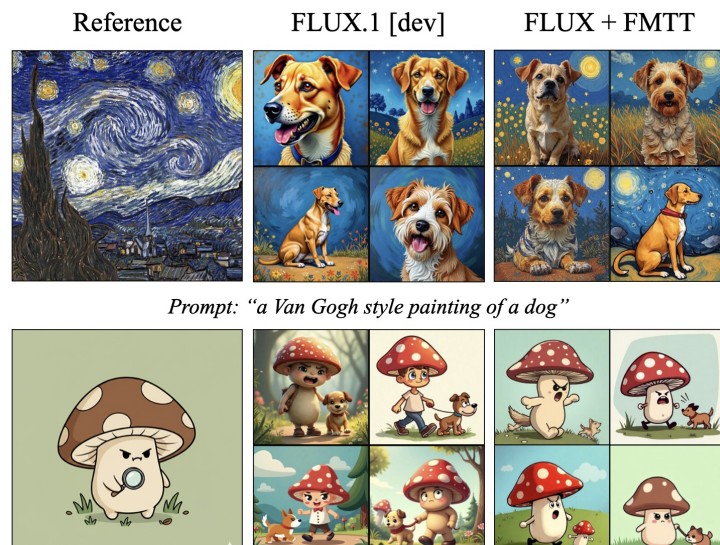

*Prompt: "a Van Gogh style painting of a dog"*

*Prompt: "a cute grumpy cartoon mushroom character walking his dog"*

**Figure 9:** Style consistency via VLM-based rewards. Given a reference image, FMTT produces images that better match its art style than the base model.

We explored two solutions. The first is to craft the VLM prompt to be as verbose and unambiguous as possible, explicitly discouraging potential "cheats". This works when only a few edge cases exist, but becomes brittle when many (4+) conditions are needed, at which point the reward model grows opaque and the search converges to local maxima. The second approach is to decompose the binary question into several simpler sub-questions and define the reward as their sum. While this adds computational overhead by requiring multiple VLM inferences, it proved more robust in practice.

For reference, to achieve the results in Figure 1, we used the following three questions:

- Is the hour hand pointing between 4 and 5?

- Is the minute hand pointing at 9?

- Is the second hand pointing at 12?

## G    ADDITIONAL QUANTITATIVE EXPERIMENTS

In this section, we further evaluate our method on the UniGenBench++ (Wang et al., 2025b) benchmark using the Skywork-VL reward model. This benchmark contains 600 short English prompts, each requiring 4 generated images. The resulting images are scored by the UniGenBench evaluation model, a finetuned variant of Qwen2.5-VL-72B (Bai et al., 2025), across multiple dimensions including entity layout, text rendering, world knowledge, and more. We present a summary of the results as a performance-vs-compute scaling figure in Figure 11 and provide the full metric-by-metric breakdown in Table 2.

As shown in Figure 11, when comparing the flow-map lookahead, the 1-step denoiser, and standard best-of-$N$ sampling, the denoiser lookahead offers little to no improvement over the best-of-$N$ baseline. In contrast, the flow-map lookahead consistently provides larger gains at lower computational cost. We attribute the denoiser's weak performance to its heavily blurred predictions at early, high-noise timesteps, which fall far off the data manifold and yield uninformative reward gradients. Figure 12 visualizes the different lookahead strategies qualitatively.

**Table 2:** Quantitative results on UniGenBench++. Entries show the percentage of images judged by the evaluation model to satisfy each criterion. The standard deviation is computed across the 4 images generated for every prompt.

| Method | Mean | Style | World Knowledge | Attribute | Action | Relationship | Compound | Grammar | Logical Reasoning | Entity Layout | Text Generation | NFE |
|---|---|---|---|---|---|---|---|---|---|---|---|---|
| **Diffusions** | | | | | | | | | | | | |
| FLUX.1 [dev] | 61.78 ± 3.58 | 85.20 ± 2.30 | 86.23 ± 2.62 | 64.53 ± 2.82 | 60.46 ± 3.80 | 63.07 ± 5.46 | 41.24 ± 5.33 | 60.96 ± 2.95 | 24.08 ± 3.63 | 67.16 ± 4.32 | 30.17 ± 5.77 | 180 |
| **Gradient-Free Search** | | | | | | | | | | | | |
| FLUX.1 [dev] + Best-of-$N$ | | | | | | | | | | | | |
| - $N$ = 8 | 65.40 ± 2.56 | 87.40 ± 2.09 | 89.72 ± 1.75 | 68.27 ± 2.50 | 62.26 ± 3.64 | 68.02 ± 4.38 | 47.29 ± 1.80 | 63.37 ± 2.16 | 29.36 ± 3.24 | 71.08 ± 71.08 | 35.06 ± 5.66 | 360 |
| - $N$ = 16 | 67.57 ± 1.53 | 88.60 ± 1.40 | 90.82 ± 1.30 | 69.44 ± 1.37 | 64.92 ± 2.33 | 70.81 ± 3.10 | 52.71 ± 2.48 | 64.97 ± 1.10 | 31.19 ± 2.34 | 72.95 ± 2.26 | 36.49 ± 4.01 | 720 |
| - $N$ = 32 | 69.18 ± 0.87 | 89.80 ± 1.40 | 90.82 ± 1.30 | 72.33 ± 1.22 | 66.06 ± 1.30 | 72.97 ± 1.98 | 54.64 ± 1.59 | 65.24 ± 1.65 | 31.88 ± 1.00 | 76.49 ± 1.54 | 39.37 ± 4.55 | 1440 |
| - $N$ = 64 | 70.93 ± 1.38 | 90.50 ± 0.59 | 90.98 ± 0.82 | 74.15 ± 1.37 | 67.97 ± 0.73 | 75.00 ± 0.66 | 56.96 ± 4.13 | 65.24 ± 3.00 | 38.30 ± 4.12 | 77.99 ± 1.71 | 42.82 ± 3.29 | 2880 |
| - $N$ = 128 | 72.77 ± 1.26 | 91.40 ± 0.66 | 91.77 ± 0.45 | 75.00 ± 1.56 | 70.15 ± 1.38 | 74.24 ± 2.36 | 60.95 ± 2.20 | 68.85 ± 2.49 | 41.97 ± 2.86 | 80.60 ± 1.40 | 44.54 ± 5.71 | 5760 |
| - $N$ = 256 | 73.75 ± 1.06 | 91.10 ± 0.59 | 92.88 ± 0.52 | 76.39 ± 0.82 | 71.58 ± 1.83 | 75.51 ± 1.54 | 61.86 ± 3.44 | 69.52 ± 0.53 | 45.18 ± 3.14 | 80.97 ± 1.54 | 44.83 ± 3.54 | 11520 |
| Flow Map + Multi-best-of-$N$ (10 rounds) | | | | | | | | | | | | |
| - $N$ = 8 | 67.58 ± 0.56 | 86.70 ± 0.77 | 90.66 ± 1.22 | 70.41 ± 1.06 | 67.40 ± 1.87 | 71.32 ± 2.05 | 52.32 ± 1.39 | 64.71 ± 2.11 | 38.30 ± 3.46 | 76.49 ± 1.63 | 18.39 ± 0.81 | 320 |
| - $N$ = 16 | 68.01 ± 1.05 | 85.90 ± 1.80 | 88.29 ± 1.05 | 72.33 ± 1.97 | 67.97 ± 1.24 | 71.57 ± 1.52 | 52.84 ± 2.30 | 67.38 ± 2.30 | 38.76 ± 3.14 | 77.05 ± 1.43 | 18.10 ± 2.35 | 640 |
| - $N$ = 32 | 70.53 ± 0.80 | 89.20 ± 0.49 | 91.14 ± 1.85 | 74.04 ± 1.67 | 70.06 ± 2.99 | 73.73 ± 2.57 | 55.93 ± 0.85 | 70.32 ± 1.79 | 39.22 ± 1.19 | 79.29 ± 2.44 | 22.99 ± 2.93 | 1280 |
| - $N$ = 64 | 70.90 ± 0.77 | 88.80 ± 1.67 | 92.56 ± 2.21 | 73.18 ± 1.49 | 70.63 ± 1.53 | 73.98 ± 2.34 | 57.73 ± 2.80 | 69.79 ± 3.04 | 41.74 ± 2.47 | 79.29 ± 3.23 | 23.28 ± 2.21 | 2560 |
| - $N$ = 128 | 72.60 ± 0.63 | 90.40 ± 1.50 | 91.46 ± 0.32 | 75.75 ± 1.33 | 70.91 ± 0.87 | 79.70 ± 2.06 | 60.57 ± 1.73 | 71.93 ± 1.79 | 42.89 ± 3.00 | 78.73 ± 2.87 | 23.85 ± 1.49 | 5120 |
| - $N$ = 256 | 73.47 ± 0.36 | 90.00 ± 1.47 | 93.04 ± 1.17 | 77.35 ± 1.24 | 75.00 ± 2.55 | 76.27 ± 0.92 | 61.73 ± 2.16 | 72.46 ± 2.16 | 46.56 ± 3.00 | 79.66 ± 2.14 | 21.55 ± 2.97 | 10240 |
| **Gradient-Based Search** | | | | | | | | | | | | |
| FMTT - 1-step denoiser lookahead | | | | | | | | | | | | |
| - $N$ = 4 | 62.00 ± 4.93 | 85.00 ± 3.26 | 84.02 ± 4.79 | 66.88 ± 4.63 | 60.08 ± 4.79 | 63.96 ± 7.57 | 42.91 ± 5.79 | 58.96 ± 3.99 | 24.77 ± 5.31 | 69.59 ± 7.13 | 28.16 ± 6.32 | 180 |
| - $N$ = 8 | 65.73 ± 1.83 | 87.70 ± 1.21 | 88.29 ± 3.30 | 69.12 ± 2.27 | 62.83 ± 1.33 | 69.04 ± 2.24 | 47.81 ± 3.54 | 62.83 ± 3.04 | 30.05 ± 2.78 | 72.57 ± 0.97 | 34.20 ± 1.70 | 360 |
| - $N$ = 16 | 67.75 ± 1.82 | 88.20 ± 0.87 | 88.92 ± 1.14 | 70.09 ± 2.16 | 65.49 ± 1.33 | 71.70 ± 2.92 | 49.23 ± 3.45 | 64.57 ± 3.77 | 32.57 ± 3.52 | 76.49 ± 3.05 | 40.80 ± 4.34 | 720 |
| - $N$ = 32 | 69.79 ± 1.06 | 90.00 ± 1.30 | 90.51 ± 1.55 | 71.90 ± 2.35 | 67.30 ± 1.17 | 73.35 ± 1.16 | 52.71 ± 1.56 | 67.38 ± 1.81 | 37.84 ± 5.44 | 77.05 ± 2.07 | 39.94 ± 1.49 | 1440 |
| - $N$ = 64 | 70.89 ± 1.32 | 90.20 ± 1.54 | 90.66 ± 1.51 | 73.72 ± 0.88 | 68.92 ± 2.89 | 73.73 ± 2.60 | 56.06 ± 1.80 | 66.58 ± 1.58 | 37.61 ± 4.05 | 78.36 ± 3.03 | 43.97 ± 2.21 | 2880 |
| - $N$ = 128 | 72.52 ± 1.42 | 91.60 ± 0.85 | 92.56 ± 0.27 | 73.40 ± 2.82 | 69.49 ± 2.75 | 76.27 ± 1.58 | 58.12 ± 2.92 | 69.52 ± 2.30 | 42.20 ± 5.07 | 81.53 ± 2.55 | 42.24 ± 0.95 | 5760 |
| - $N$ = 256 | 72.90 ± 0.89 | 91.00 ± 1.31 | 90.51 ± 1.85 | 73.93 ± 1.90 | 70.72 ± 1.57 | 78.30 ± 0.91 | 60.31 ± 0.82 | 69.65 ± 3.37 | 42.89 ± 4.12 | 81.53 ± 1.70 | 39.94 ± 1.88 | 11520 |
| FMTT - 1-step flow map lookahead | | | | | | | | | | | | |
| - $N$ = 16 | 63.28 ± 0.77 | 85.10 ± 0.95 | 85.28 ± 1.57 | 67.09 ± 2.58 | 60.93 ± 1.55 | 67.13 ± 1.31 | 45.36 ± 3.75 | 61.90 ± 1.33 | 27.06 ± 1.38 | 69.22 ± 1.43 | 27.87 ± 3.39 | 240 |
| - $N$ = 32 | 67.55 ± 0.86 | 88.60 ± 0.82 | 88.92 ± 1.05 | 68.06 ± 1.69 | 65.97 ± 1.46 | 70.94 ± 0.83 | 51.93 ± 1.17 | 65.37 ± 0.95 | 31.65 ± 3.70 | 75.37 ± 3.34 | 36.49 ± 4.33 | 480 |
| - $N$ = 64 | 71.13 ± 0.62 | 89.60 ± 0.63 | 91.14 ± 0.45 | 73.40 ± 2.17 | 67.87 ± 2.26 | 75.25 ± 1.98 | 57.35 ± 1.52 | 68.72 ± 1.34 | 37.39 ± 3.39 | 77.05 ± 2.66 | 45.11 ± 3.39 | 960 |
| - $N$ = 128 | 73.32 ± 0.68 | 90.60 ± 0.82 | 90.98 ± 0.94 | 76.39 ± 2.01 | 71.29 ± 1.46 | 75.63 ± 1.61 | 59.79 ± 1.67 | 69.65 ± 0.95 | 42.43 ± 1.76 | 80.04 ± 1.93 | 46.84 ± 2.86 | 1920 |
| - $N$ = 256 | 74.86 ± 0.50 | 91.90 ± 0.77 | 92.72 ± 1.30 | 77.78 ± 1.89 | 71.86 ± 0.60 | 77.41 ± 2.17 | 62.50 ± 1.76 | 72.59 ± 1.53 | 46.33 ± 1.38 | 81.72 ± 2.14 | 46.55 ± 4.10 | 3840 |
| FMTT - 2-step flow map lookahead | | | | | | | | | | | | |
| - $N$ = 16 | 64.73 ± 0.74 | 87.40 ± 2.09 | 87.34 ± 2.15 | 65.60 ± 2.35 | 61.79 ± 0.83 | 68.27 ± 1.81 | 47.94 ± 1.63 | 61.90 ± 1.38 | 30.50 ± 0.76 | 71.27 ± 3.10 | 33.33 ± 2.15 | 360 |
| - $N$ = 32 | 68.04 ± 1.34 | 89.60 ± 0.57 | 90.35 ± 0.82 | 70.19 ± 2.55 | 64.83 ± 4.10 | 69.80 ± 1.16 | 51.80 ± 1.06 | 66.84 ± 2.33 | 33.26 ± 4.27 | 73.69 ± 2.50 | 39.08 ± 3.45 | 840 |
| - $N$ = 64 | 71.46 ± 0.40 | 90.10 ± 1.07 | 91.46 ± 2.12 | 74.15 ± 1.11 | 68.25 ± 2.49 | 74.49 ± 1.50 | 56.70 ± 2.03 | 69.65 ± 1.28 | 38.76 ± 1.88 | 77.99 ± 2.87 | 44.83 ± 1.63 | 1680 |
| - $N$ = 128 | 74.31 ± 0.78 | 91.20 ± 1.20 | 92.56 ± 0.82 | 74.79 ± 2.28 | 71.48 ± 2.81 | 78.05 ± 0.55 | 62.89 ± 1.26 | 72.46 ± 1.26 | 46.33 ± 2.00 | 80.78 ± 2.32 | 45.98 ± 2.44 | 3360 |
| - $N$ = 256 | 75.55 ± 0.53 | 91.90 ± 0.77 | 92.88 ± 1.51 | 78.21 ± 1.09 | 72.34 ± 1.12 | 79.19 ± 1.87 | 65.08 ± 1.98 | 72.73 ± 0.85 | 46.79 ± 1.45 | 81.53 ± 1.10 | 47.70 ± 3.09 | 6720 |
| FMTT - 4-step flow map lookahead | | | | | | | | | | | | |
| - $N$ = 16 | 65.90 ± 0.45 | 86.90 ± 1.21 | 84.34 ± 1.64 | 68.27 ± 1.55 | 63.97 ± 1.43 | 69.92 ± 1.81 | 49.23 ± 1.65 | 63.90 ± 0.96 | 31.42 ± 1.76 | 75.37 ± 0.53 | 32.47 ± 1.25 | 600 |
| - $N$ = 32 | 68.12 ± 0.62 | 89.00 ± 1.00 | 87.03 ± 2.22 | 70.51 ± 1.17 | 65.02 ± 2.56 | 71.07 ± 1.83 | 52.19 ± 1.33 | 66.84 ± 3.08 | 34.17 ± 1.00 | 74.81 ± 2.14 | 40.52 ± 4.25 | 1560 |
| - $N$ = 64 | 71.48 ± 0.29 | 89.80 ± 0.72 | 89.82 ± 0.68 | 72.86 ± 0.48 | 69.39 ± 0.69 | 74.49 ± 2.19 | 56.83 ± 2.20 | 69.12 ± 0.95 | 40.14 ± 2.78 | 78.92 ± 0.97 | 43.97 ± 2.05 | 3120 |
| - $N$ = 128 | 74.32 ± 0.32 | 90.90 ± 0.71 | 90.55 ± 0.78 | 78.31 ± 0.56 | 70.63 ± 0.78 | 76.65 ± 1.19 | 60.44 ± 1.72 | 74.20 ± 1.38 | 46.56 ± 3.00 | 80.22 ± 0.83 | 49.43 ± 2.82 | 6240 |
| - $N$ = 256 | 75.14 ± 0.80 | 91.40 ± 1.18 | 92.41 ± 1.18 | 77.03 ± 1.72 | 71.39 ± 2.16 | 79.19 ± 1.48 | 65.08 ± 3.29 | 72.46 ± 2.32 | 45.87 ± 2.25 | 80.41 ± 1.10 | 50.86 ± 2.62 | 12480 |

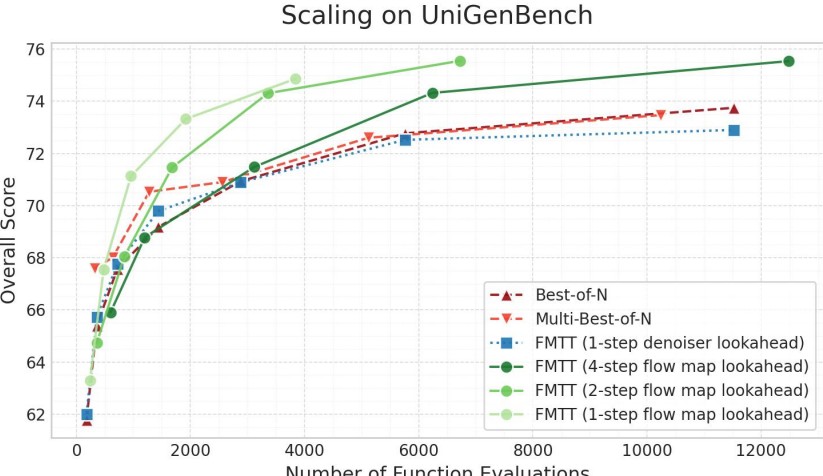

**Figure 11:** Scaling results on UniGenBench++ showing overall score versus compute. FMTT significantly outperforms both Best-of-$N$ and Multi-Best-of-$N$, achieving higher scores at comparable NFEs. Note that FMTT with a 1-step denoiser lookahead fails in beating the Best-of-$N$ baseline, demonstrating the unhelpfulness of reward gradients at blurry denoised states.

## LLM USAGE

In preparing this paper, we used large language models (LLMs) as assistive tools. Specifically, LLMs were used for (i) editing and polishing the text for clarity and readability, and (ii) generating some reference images that appear in some figures. All research ideas, experiments, and analysis were conducted by the authors. The authors take full responsibility for the content of this paper.

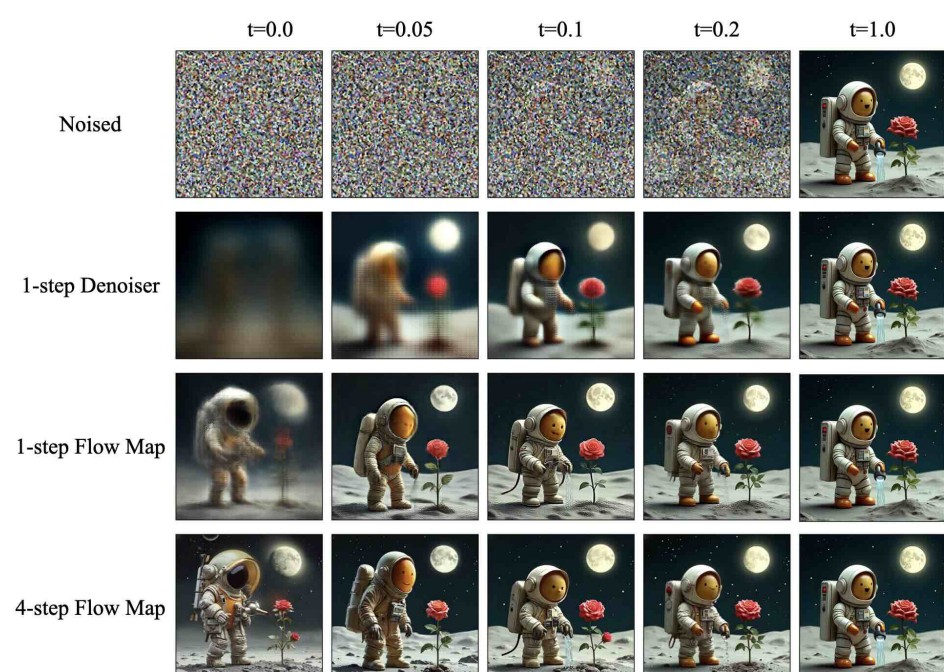

Figure 12: Comparing different lookahead methods. We visualize intermediate states for different levels of noise $t$ and show the outputs of a 1-step denoiser, a 1-step flow map, and a 4-step flow map.

