# OpenReview forum: "Test-time scaling of diffusions with flow maps"
_ICLR.cc/2026/Conference — Submitted to ICLR 2026_

### Official Review · Reviewer_84bA · 2025-10-21

**Soundness:** 3
**Presentation:** 3
**Contribution:** 3
**Rating:** 6
**Confidence:** 3

**Summary:**

This paper introduces Flow Map Trajectory Tilting (FMTT), a novel algorithm for guiding diffusion models at test-time to generate samples that align with user-specified rewards. The core problem addressed is that rewards are often only well-defined for clean, fully generated data, making it difficult to apply reward gradients during the noisy intermediate steps of the diffusion process. Traditional methods approximate the final output using a denoiser, which is often inaccurate, especially in the early stages of generation.
The key contributions of this work are:
1. It proposes using a flow map for a precise "lookahead" capability, allowing the model to accurately predict the final output from any intermediate noisy state. This solves the ill-posed nature of applying rewards mid-generation.
2. It develops the FMTT, which integrates the gradient of the reward (evaluated on the accurately predicted future state) into the generative SDE, provably performing better ascent on the reward than standard methods.
3. The method provides a principled way to perform either extract sampling of the reward-tilted distribution via importance weighting or effective search for high-reward samples.
4. It demonstrates state-of-the-art performance on complex image generation tasks that require precise control and is the first work to successfully use pretrained VLMs as complex, language-defined reward functions for test-time guidance.

**Strengths:**

This paper introduces a novel method called Flow Map Trajectory Tilting (FMTT) to address the core challenge of guiding diffusion models at test-time to generate samples that better align with a user-specified reward. Its primary innovation is replacing the heuristic "look-ahead" of traditional methods, which rely on one-step denoisers, with an "exact look-ahead" provided by a flow map. This allows the model to predict the final generated output at any point during the trajectory, providing a much stronger and more accurate signal for the reward function, especially in the early, noisy stages of generation. The approach is theoretically principled, provably performing better reward ascent than standard techniques while also enabling unbiased, exact sampling of the target distribution through a simplified importance weighting scheme.

The method is validated with high-quality and extensive empirical results. Experiments demonstrate that FMTT successfully generates images satisfying complex geometric constraints, such as symmetry and rotation invariance, where strong baseline methods fail. On the standard GenEval benchmark, FMTT achieves state-of-the-art performance, outperforming previous gradient-free and gradient-based search methods. Furthermore, the paper introduces "thermodynamic length" as a metric for sampling efficiency, showing that on a targeted MNIST generation task, FMTT not only achieves perfect classification accuracy but also has the lowest thermodynamic length, empirically confirming the method's efficiency.

This paper presents a highly original, significant, and well-executed contribution to the field of controllable generative models.

**Weaknesses:**

The paper is very strong, but there are a few limitations or areas that could be further explored:

Dependency on Flow Maps: The primary requirement of the proposed method is the availability of a trained flow map. While the paper successfully uses a model distilled from FLUX.1-dev, this dependency means the method is not universally applicable to any off-the-shelf diffusion model that is only trained to predict noise or velocity. The practicality and cost of training or distilling a high-quality flow map for other architectures could be a potential barrier to adoption.

Computational Overhead of Search: The experiments demonstrate superior performance but could benefit from a more direct analysis of the trade-off between computational budget and reward improvement.

**Questions:**

1. Could you elaborate on the sensitivity of FMTT to the quality of the flow map? How does the performance degrade if the flow map is less accurate, for instance, if it is distilled into a 1- or 2-step sampler instead of a 4-step one, or if it is trained with less data?

2. The paper demonstrates FMTT's effectiveness as a standalone guidance method. However, in many applications, other forms of "information injection" are used to control generation, such as ControlNet for spatial layout or IP-Adapters for style. Have you considered or tested the compatibility of FMTT with these methods? Could reward-based guidance from FMTT be combined with these other control mechanisms, or do you foresee potential conflicts in how they steer the generative process?

3. The experiments are based on a flow map distilled from the FLUX.1-dev model. Since its release, improved architectures within the same family, such as HiDream, have been developed. How do you expect FMTT would perform when applied to these more advanced base models? Is the effectiveness of the flow-map lookahead tied to specific properties of the original FLUX.1-dev, or would the benefits of FMTT be broadly applicable and potentially even enhanced when paired with a stronger foundation model?

---

> ### Author Response · Authors · 2025-11-25
> **Response to reviewer 84bA**
>
> We are happy to hear you found this to be a theoretically justified and empirically strong submission, and thanks kindly for your helpful feedback. Below, we detail responses to your questions, as well as new quantitative results which also help to solidify the take-home message of the paper.
>
>
> **Applicability to standard diffusion models**
>
> - Our method requires a flow map, but we see this as an opportunity rather than an obstacle. Indeed, over the past year, it has become clearer how to learn the flow map directly[1,2,3,4], and the flow map is a more general object that gives access to both the flow/diffusion model *and* the any-step model. Given that it is possible to train these now, one of the contributions of this paper is an approach to the question: what does this generalized tool make possible?
>
>
> **Computational Overhead of Search**
> - Thanks for this suggestion. We now provide a scaling study of how much compute you need to get performative results, and benchmark this scaling against the alternative methods (best-of-N search, using the denosier, etc), on an additional benchmark called UniGenBench++[1]. We chose this benchmark because, unlike GenEval, it is far from saturated (FLUX.1 [dev] scores 61.3 overall, while GPT-4o reaches 92.77), making it a more appropriate testbed for evaluation. FMTT shows both superior performance and better compute scaling compared to the ablated alternatives. We attribute this to the fact that a) this is more than search, because it uses the gradient of the reward on the fly to guide the process, and b) uses this reward how it was meant to be used -- evaluated on the data distribution. This is why we see the denoiser is no better than the best-of-N searches, as it does not provide meaningful reward signal on nuanced rewards, such as VLM-based ones. These results are included in Table 2 and Figure 11.
> - We have currently placed the new results in the Appendix, but we will reorganize the paper and incorporate the key findings into the main text for the camera-ready version.
>
> **Sensitivity to flow map performance**
> - Thanks for this suggestion. We also include the 1- and 2-step map outputs on the scaling plot in Figure 11. We observe that FMTT is quite robust to the approximation errors introduced by using a less accurate flow map, with the 1-step variant far exceeding the 1-step denoiser and best-of-N baselines in terms of both performance and computational efficiency.
>
> **Compatibility with other control paradigms**
> - We don't see any obstacle to combining this with other control paradigms that people already use with reward guidance. The only requirement would be training a compatible flow map. We leave this exploration to future work.
>
>
> **Utility beyond flux**
> - The method is applicable with any flow map. We just used a flow map distilled from FLUX due to it being a widely popular open-source model. And using a more powerful base model will only improve the results further.
>
> We hope we were able to address your questions. In particular, we hope that we could address the critical concern regarding the lack of performance-vs-compute analysis. We will include the above results and discussions in the final version of the paper. If you found our explanations convincing, we would like to kindly ask you to consider raising your score accordingly. Thank you.
>
>
> [1] Boffi, Nicholas M., Michael S. Albergo, and Eric Vanden-Eijnden. "Flow map matching." arXiv preprint arXiv:2406.07507 2.3 (2024): 9.
>
> [2] Geng, Zhengyang, et al. "Mean flows for one-step generative modeling." arXiv preprint arXiv:2505.13447 (2025).
>
> [3] Boffi, Nicholas M., Michael S. Albergo, and Eric Vanden-Eijnden. "How to build a consistency model: Learning flow maps via self-distillation." arXiv preprint arXiv:2505.18825 (2025).
>
> [4] Sabour, Amirmojtaba, Sanja Fidler, and Karsten Kreis. "Align Your Flow: Scaling Continuous-Time Flow Map Distillation." arXiv preprint arXiv:2506.14603 (2025).
>
> [5] Wang, Yibin, et al. "UniGenBench++: A Unified Semantic Evaluation Benchmark for Text-to-Image Generation." arXiv preprint arXiv:2510.18701 (2025).

---

### Official Review · Reviewer_hwMk · 2025-10-30

**Soundness:** 2
**Presentation:** 2
**Contribution:** 2
**Rating:** 0
**Confidence:** 4

**Summary:**

The paper proposes Flow Map Trajectory Tilting (FMTT), an inference-time (test-time) guidance procedure for diffusion / flow-based generative models. The core claim is that using an estimated flow map X_{t, 1} to "look ahead" to the terminal sample allows you to (1) inject reward gradients more meaningfully throughout the entire trajectory (even at early timesteps), and (2) construct importance weights whose form is allegedly simpler and theoretically grounded by a Jarzynski-style argument. The paper further claims that this enables both unbiased sampling from a tilted distribution and practical search for high-reward generations, including rewards defined by VLMs.

**Strengths:**

I think the idea is straightforward to follow. And the proof process is correct and easy to follow ( I checked all the derivations in section 2 apart from proposition 2.1 & 2.2).

**Weaknesses:**

1. The method is mostly an application of known SMC / importance weighting ideas to reward-guided diffusion, swapping in a learned flow map to get a better guess of the final clean sample. The paper over-markets this as fundamentally new.

2. The theoretical section restates standard Jarzynski/SMC logic but does not analyze estimator variance, practical degeneracy, or approximation error in the learned flow map — which are the actual hard problems.

3. The paper leans on buzzwords (“test-time scaling,” “thermodynamic length”) and selective qualitative figures instead of giving a sober, controlled, reproducible evaluation.

4. The utilization of the flow map is already a not-fresh idea at all. We have seen a bunch of work using a similar idea, such as mean flow and consistency models.

5. It lacks discussion about the motivation and intuitions of the methodology. Considering the fact that the discussed "test-time adaptation of diffusions" is already proposed by other work, the actual contributions of this work are just introducing the Flow map look-ahead and the bias correcting via importance weighting w.r.t. diffusion paths.

5. Empirical evaluation is not convincing at all.

    The experiments are positioned as if they demonstrate strong practical wins. In reality, they’re narrow, sometimes undercontrolled, and in places close to cherry-picked.

        a. MNIST “tilt to zeros”. The MNIST experiment is extremely weak as evidence. They “tilt” an unconditional model to make it generate images classified as the digit “0”, then report perfect classification accuracy and nicer thermodynamic length for their method. But MNIST digit steering with classifier gradients is a toy problem that nearly any conditional guidance trick will solve; it does not stress high-dimensional or semantic alignment. It is not credible evidence for claims about modern text-to-image alignment or test-time scaling in large vision-language reward scenarios.

        b. GenEval and human-preference rewards. The gains they claim for FMTT over strong selection baselines (best-of-N, multi-best-of-N, beam search) are marginal and sometimes inconsistent across sub-metrics. In several columns, multi-best-of-N or beam search are already competitive. The paper itself admits that FLUX.1-dev is already post-trained on human preference data, so there's limited headroom. This basically undercuts their own main quantitative table: if your headline benchmark is already saturated, then it's not a good benchmark to demonstrate superiority.

        c. There is no cost-quality frontier analysis. The method is pitched as “test-time scaling,” i.e. spend more inference compute to get better reward. But the paper does not plot reward vs. NFEs (function evals) for all baselines. Instead, it drops an NFE column in the table, but does not argue Pareto optimality. For instance, best-of-N with enough samples obviously improves, but they don't show how many samples FMTT effectively needs to match that reward level. Without compute-normalized curves, it's impossible to judge whether the method is actually more compute-efficient than naive sampling + selection.

        d. The ablations (“FMTT - 1-step denoiser lookahead”, “FMTT - 4-step diffusion lookahead”) are potentially interesting, but they’re only reported as single numbers, with no statistical significance, no variance bars, and no visibility into failure cases. We don’t see if flow-map lookahead is robust across prompts or just happened to help on a subset of lucky prompts.

In a nutshell, empirically, the paper leans heavily on selective visuals and small, convenience-scale tasks. It does not provide rigorous, large-scale, statistically grounded evidence that FMTT is (a) reliably better, (b) more compute-efficient, or (c) more robust than strong existing inference-time steering / best-of-N search baselines.

**Questions:**

Indeed, another series of work that I am more familiar with is "diffusion finetuning" (https://arxiv.org/html/2510.02692v1). I think this work's conception, "test-time scalling diffusion" is very similar to the "diffusion finetuning". Can the author explain the difference between these two lines of work? If there is no difference, can the author explain why they adopt this title instead of "diffusion finetuning"? I suspect the authors are overclaiming just to force a connection to “test-time scaling,” and I really dislike this kind of behavior. I think the bias of the sampling process with the reward terms introduced is not a fresh conclusion at all from the perspective of "diffusion finetuning".

Besides, for the visualization, indeed, I have tried myself with those prompts provided by authors. I found that, at least for GPT5, it can work well for those prompts shown in figure 3. The clock prompt in Figure 1 is indeed problematic for GPT5.

I think the author should also conduct a detailed ablation study about the precision regarding the guidance as well as the generation quality.

There are three typos I found:

1. line 191 "so that so that".

2. line 312, "proposition 3.2"

3. line 312, "proposition" lacks a hyperlink.

---

> ### Author Response · Authors · 2025-11-25
> **Response to Reviewer hwMk**
>
> We thank Reviewer hwMk for their detailed feedback and acknowledge several legitimate points that we will address below in our response. However, we respectfully note that the review's tone and the score of 0 (strong reject) appear disproportionate to the substance of the critiques raised, particularly given that three other reviewers assigned scores of 4, 6, and 10. Phrases such as "over-markets," "not credible evidence," and "I really dislike this kind of behavior" suggest an emotional tenor that we believe is not conducive to productive scientific discourse and not reflective of the substance of our paper. We are committed to addressing the technical and empirical concerns raised in a rigorous and objective manner, and we hope the remainder of this discussion can focus on the scientific merits and limitations of our work, free from subjective judgments about intent or character.
>
> To that end, we hope to use this rebuttal, as well as revisions suggested by the substance of your review, to show that our method is a) rigorously justified, b) a novel contribution, and c) quantitatively favorable across a suite of benchmarks, and not merely marketing. We appreciate your input and look forward to engaging with you on these points:
>
>
> **Concerns about novelty**
>
> - The reviewer argues that our paper is not novel because it relies on the use of sequential monte carlo in test-time adaptation. We would like to stress that the **point** of our paper is a) to show that standard ways people approach the question of SMC in test-time adaptation is **flawed**, and b) there is a principled way to correct that which relies non-trivially on a relation between the flow map and the instantaneous diffusion *realized simultaneously through the flow map parameterization*, and c) new rigorous justification of the approach through mathematically simplified importance weights and analysis of the associated utility of the flow map tilting through the thermodynamic length. We hope to stress that dismissing the paper wholly because it "relies on SMC" feels quite unfair, because we want to show the community how to do this line of work correctly and more efficiently by a combination of new mathematical insights and large scale experiments.
>
> **We do not analyze estimator variance, practical degeneracy, or approximation error in the learned flowmap**
> - Regarding the impact of approximation error in the learned flow map, please see Figure 11 in Appendix G, where we compare between 4-step, 2-step, and 1-step flow map lookahead, as well as a 1-step denoiser lookahead, across a range of computational budgets. Even a 1-step flow map, despite its larger approximation error, provides a clear improvement over the best-of-$N$ baseline. In contrast, the 1-step denoiser introduces much larger errors and offers little to no gain over best-of-$N$.
> - By estimator variance, we believe that the reviewer refers to the variance of the SMC normalization constant, which is a proxy for how well the SMC algorithm samples from the tilted distribution. In our work, we consider sampling algorithms indexed by the time-dependent reward $r_t$ and the function $\chi_t$ (see App. A.3). While the variance of the SMC normalization constant is a criterion to assess the best choice, it depends strongly on the discretization parameters of the dynamics (number of timesteps, number of particles, number of resampling steps), and thus does not inform the continuous time dynamics choice. In Appendix B we introduce the total discrepancy and the thermodynamic length, which are connected to the variance of the SMC normalization constant through Theorem B.1, and which are reflective of the continuous-time dynamics. To summarize, we develop theoretical tools to understand what are the best sampling algorithms when several choices are available. We would like to emphasize that these are not buzzwords or concepts that we introduce for marketing purposes.
> - We would appreciate a clarification on what the reviewer means by "practical degeneracy".
>
>
> **Flow maps have already been invented**
>
> - Indeed, there are a number of works that show how to learn flow maps and detail what equations govern this learning[1,2,3,4]. **We are not claiming to have invented flow maps** -- we are showing how mathematical relations arising from them allow for new, effective ways of performing test-time adaptation. An open research question is how to take advantage of flow maps now that they do exist. Our work is one such contribution in that regard. If you don't think this is made clear in the intro, please let us know.

---

> > ### Author Response · Authors · 2025-11-25
> > **Response to reviewer hwMk, part 2**
> >
> > **Concerns regarding the experiments**
> > - **MNIST experiments:** The MNIST experiments were not meant to serve as the quantitative evidence that the method works well -- it is merely a testbed for which we could validate the theoretical motivation for the paper, at a scale at which the theoretical quantities of interest (e.g. thermodynamic length) were actually computable (otherwise they are too expensive). Given that the main experimental table was only from GenEval and these are saturated, we understand reading it this way. However, the purpose it was meant to serve was "let's engage with the theory at the experimental scale at which it is possible to get a sense of things, and, from there, we'll see what happens in the wild at really large scale". As such, we push back against the notion that these are just "buzzwords" as they are the motivation for how we arrived at the approach.
> > - **UniGenBench++ experiments** We agree more experimental validation beyond GenEval would be useful, especially quantitative scaling studies. To address this, we ran a new set of experiments using the Skywork-VL VLM-based reward and on the UniGenBench++[5] benchmark. This benchmark measures performance across a variety of dimensions such as world knowledge, logical reasoning, and spatial layout. We chose this benchmark because, unlike GenEval, it is far from saturated (FLUX.1 [dev] scores 61.3 overall, while GPT-4o reaches 92.77), making it a more appropriate testbed for evaluation. The results are presented as a performance-vs-compute plot in Figure 11, with a full breakdown in Table 2. We observe that FMTT consistently outperforms the baseline methods, such as best-of-N and 1-step denoiser lookahead, while using far fewer function evaluations. Additionally, the plot makes clear how the gradient obtained by a 1-step denoiser lookahead is completely non-helpful and even detrimental to the overall performance at extremely large scales. This supports our main claim: when a complex reward is computed on an overly smoothed 1-step denoised state, it provides almost no useful gradient signal and offers little advantage over standard sampling.
> >
> >
> > **Ablations missing variance estimates**
> > - We have added standard deviations to Table2 to demonstrate spread, as this is the approach taken in the paper you linked to (https://www.arxiv.org/pdf/2510.02692). We will make sure to also add the standard deviations to Table 1 for the camera-ready version. Also, please note that this paper was posted *after* the ICLR deadline, and applies to fine-tuning (i.e. learning) and not inference time adaptation.
> >
> > **Discussion on motivation and methodology**
> > - We stress that the motivation and methodology of this paper relies on an insight connecting flow maps and instantaneous velocity fields. Because one model contains both the instantaneous transport and the one-step model, we can use a **single model to properly make use of rewards** in test-time scaling of the diffusion sampler. In addition, doing so simplifies the importance weights, and improves their variance (thermodynamic length is shorter). Of course, there are other papers on test-time scaling of diffusions. The point of our paper is to say "If you're going to do that, this is the proper way to handle it". If this is not clear, please let us know.
> >
> >
> > **Fine-tuning vs test-time adaptation**
> > - Fine-tuning models and test-time scaling are very similar in their motivation, which is to generate outputs that score higher on some specified reward function. However, when doing finetuning, the parameters of the model are optimized such that standard diffusion sampling with the new model will result in higher-reward outputs. This is contrast to test-time scaling methods, which assume the base model to be fixed, and attempt to generate high reward outputs via search or guidance. Because of this distinction, the two are treated as separate lines of work; our paper contributes to the latter.
> > - Fine-tuning also requires the reward function to be fixed. If the reward changes, the model must be retrained. Test-time scaling avoids this constraint, allowing the reward to be modified freely without altering the underlying algorithm. This flexibility is what enables us to switch between very different VLM-based rewards (style consistency, character consistency, text readability, etc.) simply by changing the yes/no question being asked. Examples are shown in Figures 1 and 9 of the paper.
> >
> > We hope these amendments and clarifications make clear that the points you wanted us to quantitatively show are true: a) We have statistically grounded evidence that FMTT is reliably better b) more compute-efficient and c) more robust than best-of-N search baselines. Given these clarifications, we would be grateful if you could reconsider your score under the evaluations provided herein.

---

> > > ### Author Response · Authors · 2025-11-25
> > > **response to reviewer hwMk, part 3**
> > >
> > > **Citations**
> > >
> > >
> > > [1] Boffi, Nicholas M., Michael S. Albergo, and Eric Vanden-Eijnden. "Flow map matching." arXiv preprint arXiv:2406.07507 2.3 (2024): 9.
> > >
> > > [2] Geng, Zhengyang, et al. "Mean flows for one-step generative modeling." arXiv preprint arXiv:2505.13447 (2025).
> > >
> > > [3] Boffi, Nicholas M., Michael S. Albergo, and Eric Vanden-Eijnden. "How to build a consistency model: Learning flow maps via self-distillation." arXiv preprint arXiv:2505.18825 (2025).
> > >
> > > [4] Sabour, Amirmojtaba, Sanja Fidler, and Karsten Kreis. "Align Your Flow: Scaling Continuous-Time Flow Map Distillation." arXiv preprint arXiv:2506.14603 (2025).
> > >
> > > [5] Wang, Yibin, et al. "UniGenBench++: A Unified Semantic Evaluation Benchmark for Text-to-Image Generation." arXiv preprint arXiv:2510.18701 (2025).

---

> ### Comment · Reviewer_sdVa · 2025-11-27
>
> It is very appreciated that the reviewer hwMk checked the proofs in detail. However the authors have a point that assumptions about character and intent must be left out of the scientific discussion. I'd just like to add a few more points that I find deviate from the high standards that a strongly-opinionated review must be held to:
>
> It is too vague to say "We have seen a bunch of work using a similar idea, such as mean flow and consistency models" without citations and without bridging the concepts in both works at a technical level -- if it's so similar, it should be easy to make this connection, and very specifically.
>
> Also, claiming that this work is very similar to "diffusion finetuning" while in the same phrase asking the authors to explain the differences undermines the point. If the reviewer hwMk believes this, it must be strongly supported in a good understanding of both works, so a generic explanation by the authors is not needed. If the reviewer has specific points to ask about then this would be of course legitimate.

---

### Official Review · Reviewer_sdVa · 2025-10-31

**Soundness:** 4
**Presentation:** 4
**Contribution:** 4
**Rating:** 10
**Confidence:** 3

**Summary:**

This paper proposes a method to improve test-time sampling of diffusion models according to arbitrary rewards. The method is based on flow maps, which have two time arguments and can be used to "jump" to approximations of the solution rather than require full denoising, and this is leveraged to apply the reward model to denoised samples, rather than noisy ones (which would be out-of-distribution). Several qualitative and quantitative results are presented.

**Strengths:**

This paper seems to present a well-reasoned proof that flow maps can be used to apply rewards to diffusion models with look-ahead, so that the rewards are more accurate than if they were applied to the noisy samples near t=0. While I did not check all the technical details, I did not find any errors in the math, and it makes sense. The experimental results are excellent, and show multiple creative uses of the rewards, especially in ways that are out of distribution for previous benchmarks. Barring any mistakes (which would be surprising given the good experimental results), this seems like a very solid contribution to the field.

**Weaknesses:**

I did not find any major weaknesses. I only have a few suggestions for improving the clarity slightly:

- Fig. 2 needs labelling of the time axis.

- Nabla is overloaded, both as an operator and stand-alone symbol (composed with dot product); although it is common notation, due to the overloading it would be safer to define it in all cases.

- Fig. 5: It would be more convincing to present a scatter plot of thermodynamic lengths vs. reward across samples, instead of just average bar plots, which is much weaker evidence for the conclusion.

**Questions:**

I have no further questions for the authors.

---

> ### Author Response · Authors · 2025-11-25
> **Response to reviewer sdVa**
>
> We are excited to hear that you found the paper to be a performant and rigorously justified submission! Below, we detail some of the revisions and clarifications we've made based on your feedback.
>
> **Time labeling in figure 2**
> - We have amended this plot to make clear the time axis. Thank you.
>
> **Use of $\nabla$**
> - We will do our best to modify any areas where the use of $\nabla$ is opaque. Just to clarify, everywhere in the document, the gradient $\nabla$ is taken with respect to the input $x$. Because $\nabla \cdot$ is standard notation for the divergence operator (with respect to $x$), we are hesitant to change it for sake of potentially confusing others, but we will add remarks wherever it is ambiguous about which derivative we are taking.
>
>
> **Bar plots vs scatter plots**
> - We have expanded our exposition in general on the thermodynamic length (see appendix of revised paper), and added several additional plots in App. D. Because in this case there are many plots to juxtapose, we found the scatter plot a bit harder to read. Let us know what you think, and we are open to changing it if you think it'd be best.
>
>
>
> **New experimental validation and scaling costs**
> - Though you did not request them, we've added even more numerical validation of our method, showing that on a separate thorough benchmark called UniGenBench++[1], our method significantly outperforms alternatives in composite score, and does so with significantly less compute compared to the best results of the other methods. We hope you find these results interesting. They are provided in the new submission in Table 2 and Figure 11.
> - We have currently placed the new results in the Appendix, but we will reorganize the paper and incorporate the key findings into the main text for the camera-ready version.
>
>
> We hope we were able to address your questions and that you find the new results helpful. Thank you again for your thoughtful and positive review.
>
> [1] Wang, Yibin, et al. "UniGenBench++: A Unified Semantic Evaluation Benchmark for Text-to-Image Generation." arXiv preprint arXiv:2510.18701 (2025).

---

> > ### Comment · Reviewer_sdVa · 2025-11-27
> >
> > Thank you for the explanation, I agree. I cannot find the scatter plots though; Appendix D has MNIST experiments. My request was focused on the correlation between thermodynamic lengths vs. reward.

---

### Official Review · Reviewer_t4do · 2025-10-31

**Soundness:** 2
**Presentation:** 2
**Contribution:** 3
**Rating:** 4
**Confidence:** 2

**Summary:**

The author introduce Flow Map Trajectory Tilting (FMTT), a method for sampling from reward-tilted probability distributions in the setting of flow-map-based generative models.

The most widely known method for doing this in the case of diffusion models is classifier guidance, which consists of adding the gradients of a learned reward function to the score predictions of a diffusion model to guide samples towards regions of high reward. While this approach is intuitive, it needs to employ heuristics to compute rewards for noised latents, as the pre-trained reward function is trained only on noise-free samples.

The authors aim to bridge this gap by considering generative models parameterized by flow maps (i.e. maps $X_{s, t}$ associated with an ODE such that, for any solution $x$ of this ODE, $X_{s, t}(x_s) = x_t$). In this setting, at noise level $s$, an estimate of the noise-free latent can be computed as $X_{s, 1}(x_s)$ (where $t=1$ corresponds to the data distribution). The reward can then be evaluated at this estimate, and its gradient can be added to the score (which is also obtainable from the flow map).

However, directly applying this approach does not produce samples from the correct reward-tilted data distribution. Rather, the authors show the dynamics have an extra additive term that is not present in the dynamics of the true reward-tilted measure of interest. They propose correcting for this additional term through importance sampling. The importance sampling weights are obtained from the Jarzynski estimator, which takes on a simplified form in the specific case of interest. The authors then propose using Sequential Monte Carlo to sample trajectories with these importance weights, hence producing unbiased samples from the true reward-tilted measure.

Empirically, FMTT shows (i) modest GenEval gains on human-preference rewards; (ii) strong improvements on geometric/structural rewards (masking, symmetry, rotation), where flow-map look-ahead outperforms denoiser/diffusion look-ahead; (iii) the first effective use of vision-language models as rewards for natural-language alignment/editing.

**Strengths:**

- Important problem: reward conditioning in diffusion models and related generative models is a fundamental question with far-reaching implications for various fields like image and video generation, robotics and scientific applications.
- Non-trivial, theoretically principled technique: the construction (and comprehension) of FMTT demands a fair amount of sophistication in stochastic analysis, PDEs and Monte Carlo methods. As such, most researchers in the field would not have been able to arrive at these ideas on their own, meaning this paper adds counterfactual value to the field, provided the techniques’ effectiveness can be conclusively demonstrated.

**Weaknesses:**

- Unclear motivation for why flow maps are more appropriate than diffusion models for reward guidance: in section 2.2, the authors claim that passing a noisy latent $x_s$ through a denoiser $D$ before computing the reward is inappropriate due to the denoiser providing little information early on in the dynamics. It is not a priori clear to me why using the flow map $X_{s,1}$ instead would not share the same problem for small $s$. It would be good if the authors could clarify this.
- The presentation makes comprehension difficult for those not already familiar with the technical tools used in this work. For example, the section on SMC and thermodynamic length is very difficult to understand if one is not already familiar with SMC and related concepts, even having a background in diffusion models and stochastic analysis. From reading Proposition 2.3 in the main paper, it is not clear what thermodynamic length is or what intuitions it corresponds to. Similarly, it is not intuitive how one arrives at the SDE for the importance weights in Equation 21. Rephrasing these technical sections with a bigger focus on intuition would likely help technical readers who are not experts in the specific technical tools used here to better understand the method. I believe improving this will be important to ensure the broader community can understand the method and consider adopting it or building on it. It would also be good to provide background on SMC in section 2.1.
- Scope of evaluations. Many results are qualitative for specialized rewards; quantitative studies (e.g. multiple datasets, objective quantification of output quality) would strengthen claims beyond case studies. There exists literature on using diffusion for solving constraint satisfaction problems; see e.g. https://arxiv.org/abs/2211.15657 . Such setups could be adapted here to demonstrate whether FMTT leads to practical improvements in constraint satisfaction in these environments where success is likely easier to measure, compared to image generation.
- Lack of a systematic account of how performance scales with NFE: given that the paper positions itself as a test-time scaling technique, it is important to have a sense of how performance varies with the amount of compute used at inference time. NFE numbers are reported in Table 1, but a more rigorous comparison would give a better sense of e.g. whether FMTT Pareto-dominates other baselines, or whether it only performs better if significant compute is expended.

**Questions:**

See the Weaknesses section.

I consider this paper to be interesting from a theoretical and conceptual perspective, but it is , in its current form, held back (in my opinion) by unclear exposition and insufficient evaluations to establish the practical performance of FMTT relative to other approaches in a systematic, quantitative way. I believe it can be possible to address this in the rebuttal, and I am open to increasing my score if the authors extend their qualitative comparisons to quantitative ones, include performance-vs-compute scaling charts for FMTT and relevant baselines in some of the tasks considered in Table 1, and make the exposition clearer regarding motivation and SMC.

---

> ### Author Response · Authors · 2025-11-25
> **Response to reviewer t4do**
>
> Thanks to the reviewer for the very helpful comments. We are glad you found this to be a novel and precise contribution to the topic of inference-time adaptation of flows and diffusion.  We are happy to provide more experimental validation of the method, as well as added clarity to the mathematical contributions of our paper as they pertain to best ways of using the flow map, and the importance of the flow map in this setup.
>
>
> **Unclear motivation of the flow map as opposed to diffusion models**
> - Our method relies on a key insight known in the literature that having access to a flow map $X_{s,t}(x) = x + (t-s) v_{s,t}(x)$, which is the exact solution operator for an ODE $\dot x_t = b_t(x_t)$, also means we have access to the instantaneous velocity $b_t$ which solves the usual flow matching/diffusion problem. It's a more general object, so, when learned, we could exactly evaluate the one-step map $X_{0,1}(x_0) = x_1$, but we could also use the diagonal $v_{t,t}(x) = b_t(x)$ ..... via $$dx_t = v_{t,t}(x_t) dt + \epsilon_t s_t(x_t)dt + \sqrt{2 \epsilon_t} dWt $$ where $s_t$ is a re-writing of $b_t$. *By using both of these benefits simultaneously, we can exactly look ahead to what sample we are headed to produce, and evaluate the rewards on a clean sample*, that is:
> $$dx_t = v_{t,t}(x_t) dt + \epsilon_t [ s_t(x_t) + \nabla r(X_{t,1}(x_t))] + \sqrt{2 \epsilon_t} dWt $$ Here, $X_{t,1}(x_t)$ is the *exact look-ahead of what sample we would have gotten, given that we are at $x_t$ along the SDE trajectory*. This is distinctly different from just using the denoiser because the denoiser does not give you a sample $x_1 \sim \rho_1$. In particular, using the denoiser from time $0$ to time $1$ would give you a meaningless output, whereas our map $X_{0,1}(x_0) = x_1$ when learned would truly give you a sample under $\rho_1$. We show an example of the outputs of the denoiser and the flow map at different noise levels in Figure 12.
>
> **More explanation and intuition on the SMC/Thermodynamic length discussions**
>
> - Thanks for suggesting this -- we have reworked the discussion of the thermodynamic length to give it more context, and provided more information in the appendix analyzing it. In particular, in Appendix A.3 we present four different sampling algorithms (one of them the one introduced in the main text), and all of them belong to a family of algorithms indexed by a function $\chi_t$. Proposition A.1 shows that all these algorithms provably sample the tilted distribution, for any reward $r_t$ such that $r_1 = r$. Hence, we want to find a criterion to understand which choice of $\chi_t$ and $r_t$ is best. While the variance of the SMC normalization constant is a way to measure how a particular algorithm performs, it depends strongly on the discretization parameters of the dynamics (number of timesteps, number of particles, number of resampling steps), and thus does not inform the continuous time dynamics choice. In Appendix B we introduce the total discrepancy and the thermodynamic length, which are connected to the variance of the SMC normalization constant through Theorem B.1, and which are reflective of the continuous-time dynamics. To summarize, we develop theoretical tools to understand what are the best sampling algorithms when several choices are available.
>
> **More quantitative evaluations and scaling figure**
> - We agree that additional quantitative evaluations beyond GenEval are valuable, particularly a scaling analysis. To address this, we ran a new set of experiments using the Skywork-VL VLM-based reward and performed evaluation on the UniGenBench++[1] benchmark, which measures performance across diverse skills such as world knowledge, logical reasoning, and spatial layout. We chose this benchmark because, unlike GenEval, it is far from saturated (FLUX.1 [dev] scores 61.3 overall, while GPT-4o reaches 92.77), making it a more appropriate testbed to evaluation our method.
> - The results are summarized in Figure 11 with a full breakdown provided in Table 2. FMTT consistently outperforms the baseline methods while using far fewer function evaluations. Additionally, the plot shows that FMTT with a 1-step denoiser lookahead  observation from the scaling plot is that FMTT with a 1-step denoiser lookahead performs roughly on par with naive Best-of-N. This supports our intuition: when a complex reward is computed on an overly smoothed denoised state, it provides almost no useful gradient signal and offers little advantage over simple sampling.
> - We have currently placed the new results in the Appendix, but we will reorganize the paper and incorporate the key findings into the main text for the camera-ready version.

---

> > ### Author Response · Authors · 2025-11-25
> > **Response to reviewer t4do**
> >
> > - We also appreciate the reviewer pointing to constraint-satisfaction problems, as they were originally the motivation behind our geometric rewards. However, to ensure our evaluation is compatible with prior and future works, we decided to use UniGenBench++ for our additional experiments.
> >
> > We hope that you find these clarifications and additional experiments useful and sufficient. Please let us know if there is any other information we can provide to elucidate the theoretical justification and performance of our method. If not, we would like to kindly ask you to consider raising your score accordingly. Thank you.
> >
> >
> > [1] Wang, Yibin, et al. "UniGenBench++: A Unified Semantic Evaluation Benchmark for Text-to-Image Generation." arXiv preprint arXiv:2510.18701 (2025).

---

### Author Response · Authors · 2025-11-25
**General response to reviewers**

Dear all,

We sincerely appreciate your helpful feedback and are glad to see that the paper was well received by the majority of reviewers. During the rebuttal period, we conducted additional experiments and added several clarifications we would like to highlight below.

- **Additional Quantitative Experiments**: Several reviewers requested further quantitative comparisons between FMTT and prior approaches. In response, we ran new experiments using the Skywork-VL VLM-based reward and evaluated on the UniGenBench++ [1] benchmark, which measures capabilities across world knowledge, logical reasoning, spatial layout, and more. We selected this benchmark because, unlike GenEval, it is far from saturated (FLUX.1 [dev] scores 61.3 overall, while GPT-4o reaches 92.77), making it a more appropriate testbed for evaluating our method. The results are shown as a **performance-vs-compute plot** in Figure 11, with a detailed breakdown provided in Table 2. Across all computational budgets, FMTT consistently outperforms baselines such as best-of-$N$ and 1-step denoiser lookahead, while requiring significantly fewer function evaluations. The plot also illustrates that gradients derived from a 1-step denoiser lookahead are completely ineffective and even detrimental at large scales. This reinforces our main claim: when a complex reward is computed on an overly smoothed 1-step denoised state, it yields almost no useful gradient signal and offers little benefit over standard sampling. In contrast, using a flow-map lookahead avoids this issue entirely.
- **Improved Theory Exposition and Sampling Experiments**: We unified our analysis of the possible sampling algorithms by indexing them with a function $\chi_t$, with four particular choices of relevance (see App. A.3). The algorithm presented in the main text corresponds to $\chi_t \equiv 0$. In App. D we tested the four $\chi_t$ choices, with flow map look-ahead, denoiser look-ahead and no lookahead, on the MNIST setting.

We have highlighted all changes made during the rebuttal period in bright blue for easier visibility. This will be removed for the camera-ready version. We believe these additions further strengthen the submission.

Thank you for your time and feedback,

The authors of FMTT

---

### Meta-Review · Area_Chair_jVPB · 2026-01-10

**Summary:**

In my recommendation, I primarily relied on the two mid-range reviews (t4do: 4, 84bA: 6) and explicitly down-weighted the extreme scores (sdVa: 10 and hwMk: 0) as less reliable signals. The decision is mainly informed by: (i) the paper’s key claim of compute-efficient “test-time scaling” was not initially supported by sufficiently broad, compute-normalized comparisons against strong baselines; (ii) the method’s practicality and generality are constrained by requiring a high-quality flow map, with limited discussion of cost, availability, and failure modes; and (iii) important technical components (SMC/importance weighting and thermodynamic length) are hard to follow in the main text, weakening accessibility and reproducibility.

**Reviewer Concerns:**

- Addressed by the rebuttal: The authors added a new compute-versus-performance evaluation (UniGenBench++ with a scaling plot and breakdown table), which directly targets the request for compute-normalized evidence and partially strengthens the empirical case. They also provided a clearer motivation for flow-map look-ahead versus denoiser look-ahead and reported expanded discussion/analysis of thermodynamic length (mostly in appendices).

- Still outstanding: The new results improve the situation but remain limited in scope relative to the strength of the claims; broader validation across multiple datasets/reward types and more comprehensive Pareto frontiers vs strong baselines are still missing. The adoption barrier from requiring flow maps (training/distillation cost, robustness across architectures/training regimes) is still not concretely analyzed. Finally, key technical intuition and supporting evidence are not yet cleanly integrated into the main narrative (and at least one reviewer could not locate the requested correlation evidence), so clarity concerns remain.

**Reviewer Scores:**

t4do (4): likely to remain 4 (possibly 6 if the new scaling evidence is strong and clearly integrated), but remaining concerns about breadth and clarity would keep it near the borderline.

84bA (6): likely to remain 6 due to the added scaling study and sensitivity indications, but flow-map dependency and limited practicality evidence would keep it borderline.

sdVa (10, down-weighted): likely to remain ~9–10; minor reduction at most due to missing/mislocated requested analysis.

hwMk (0, down-weighted): likely to remain very low; at most a small increase (e.g., ~1–2) given the added scaling results, but not a material change in stance.

---

### Decision · Program_Chairs · 2026-01-26

Reject